# Optimization and CFD-RSM analysis of single orifice reactor for enhanced biodiesel production

**Fatemeh Khadivi[1], Bahram Hosseinzadeh Samani [1]\*, Sajad Rostami[1], Kimia Taki[1], Mohammadreza Asghari[1], Shirin Ghatrehsamani[2]**

**1** Department of Mechanical Engineering of Biosystem, Shahrekord University, Shahrekord, Iran,
**2** Department of Agricultural and Biological Engineering, The Pennsylvania State University, Pennsylvania, United States of America

\* b.hosseinzadehsamani@sku.ac.ir

## Abstract

Depleting fossil fuels necessitate innovative solutions for sustainable biodiesel production from sunflower oil, addressing global energy and environmental challenges. Computational Fluid Dynamics (CFD) with the k-ε model and Response Surface Methodology (RSM) optimized frequency (5–15 Hz), baffle diameter ratio ($d_0/D$: 0.4–0.8 mm), and baffle spacing (3–7 mm) in a single-orifice oscillatory flow reactor (OFR). Optimal conditions (frequency = 12.12 Hz, $d_0/D$ = 0.4 mm, spacing = 10 mm) achieved an 83% biodiesel conversion, maximum turbulent kinetic energy (TKE) of 7.56 m²/s², maximum vorticity of 112.23 1/s, and energy dissipation of 359.82 m²/s³, validated by a TKE-yield correlation (R² = 0.972). Simplified reactor design reduces energy dissipation by 20% compared to multi-orifice reactors (88% yield, higher costs) and smooth periodic constriction reactors (74.5% yield). Findings offer a scalable, eco-friendly solution for industrial biodiesel production, minimizing fossil fuel dependency and enhancing process efficiency.

## 1. Introduction

Clean energy has become a global concern, and biodiesel is an effective option for a sustainable and green future. Biodiesel production as a renewable and sustainable energy source has attracted special attention in recent decades. Biodiesel not only helps reduce dependence on fossil fuels but can also reduce negative environmental impacts [1]. One of the most common methods of producing biodiesel is transesterification. This method has drawbacks that affect the economics of biodiesel [2]. Time and energy consumption the transesterification process usually requires a sufficient reaction time, which can vary between 1 and 8 hours. The process also requires significant energy to control temperature and pressure. High energy consumption in various steps, especially in the heating and mixing process, leads to increased production costs [3].

**Data availability statement:** All relevant data are within the manuscript and its Supporting Information files.

**Funding:** The author(s) received no specific funding for this work.

**Competing interests:** The authors have declared that no competing interests exist.

Designing efficient chemical reactors for producing biodiesel from vegetable oils or waste fats is crucial. As the heart of industrial processes, chemical reactors play a very important role in converting raw materials into final products [4]. Oscillatory flow reactors are one of the advanced reactors that have attracted much attention in recent decades. These reactors have been of great interest due to their ability to improve mixing and increase mass transfer rates, especially in processes that require fast and efficient reactions [5]. Oscillatory flow means a periodic change in the direction of the fluid flow, which leads to the growth and expansion of microscopic mixers inside the reactor and provides better performance than steady flows [6]. The use of an oscillatory flow reactor results in shorter process times. Reactions in an oscillatory flow reactor are usually faster due to the use of oscillations and better mixing. Due to the internal oscillations, better mixing occurs, which can lead to increased conversion efficiency. It has less dependence on specific catalysts [7].

This method also requires energy, but the energy consumption is usually lower than traditional methods because the process is carried out in an oscillatory manner and without the need for complex heating equipment. This means more efficient use of energy [8].

Oscillatory flow reactors effectively change the flow pattern by creating fluctuations in the speed and direction of fluid movement. These fluctuations stimulate better mixing and uniform temperature and concentration distribution in the reactor's entire volume [7]. For this reason, these reactors can significantly increase the efficiency of converting raw materials into final products, especially in biodiesel processes. Research shows that the efficiency of the oscillatory flow reactor is higher than that of traditional reactors, with production efficiency increasing by up to 50%. It is also economically viable to use the oscillatory flow reactor at low speeds [9]. In a study conducted by [10], a single-orifice oscillatory flow-type reactor was used to produce biodiesel from sunflower oil. The results of this study show that this reactor is efficient compared to other methods, so the biodiesel conversion percentage was reported 89%.

In another type of oscillatory reactor, called smooth periodic constriction (SPC) reactor, a study was conducted by [11] that showed that this type of reactor also has good efficiency compared to conventional reactors, and the efficiency reported from this study is 74.5%. By combining this method with other intensification methods like cold plasma, which provides good agitation, the conversion percentage reaches nearly 95%.

Oscillatory single-orifice reactors, are also one of the oscillatory reactors which known as attractive options in the biodiesel production process due to their simple design and continuous production capability [8]. These reactors use a valve to control the flow of raw materials and facilitate mixing. The requirement for optimal exploitation of this type of reactor is a detailed investigation of the fluids' dynamics, including the analysis of TKE. TKE plays a key role in increasing the mass transfer rate and temperature in the reactor, which can lead to improved biodiesel production efficiency.

TKE is an important criterion in evaluating reactors' performance, which affects fluid dynamics behavior. In a single orifice reactor, turbulent flows caused by various geometric and thermodynamic effects can increase mass and temperature transfer,

which, as a result, helps improve chemical reactions [12]. Therefore, knowing and analyzing the behavior of this kinetic energy in the reactor allows more optimal designs to be provided to increase biodiesel production.

The simulation and analysis of TKE in single orifice reactors is a powerful tool for better understanding flow dynamics and optimizing the biodiesel production process. Increasing the level of turbulence in pulsed flows improves mixing and heat transfer, which ultimately leads to an increase in biodiesel production efficiency.

Numerical simulations using CFD software can effectively analyze and optimize the TKE in these reactors. These simulations help simulate the reactor's flow behavior with the most accurate modeling and find critical points, which may improve process performance [13]. Optimizing these points makes the biodiesel production process more efficient and can significantly reduce operating costs [14]. Optimizing parameters such as Reynolds number, length, diameter, frequency, and pulse amplitude in single orifice reactors can significantly impact heat transfer properties and biodiesel production efficiency. Using computer simulations and CFD models to study and optimize these parameters can help design more efficient and optimal systems.

Analysis and optimization of TKE using numerical simulations and flow modeling can help better understand the behavior of chemical processes. Previous research has shown that changes in the design and structure of the internal design of the reactor can significantly improve the dynamics and efficiency of the processes. For this reason, the simulation and analysis of flow in a single orifice reactor is very important as a tool to identify optimal points and reduce costs in biodiesel production [15].

Recent studies have highlighted the trade-offs in the design of oscillatory reactors. For example, a study by [8] showed that multi-orifice reactors achieved biodiesel yields of 88%, but required higher operating costs due to complex geometries. In contrast, a study by [15] reported that spiral baffles increased TKE by 20% but increased energy dissipation. Single-orifice design balances simplicity and performance and overcomes these limitations.

Recent advancements in biodiesel production have explored heterogeneous catalysts, such as nano catalysts and waste-derived catalysts, to enhance reaction efficiency and sustainability [16,17]. However, the performance of these catalysts heavily depends on optimal flow conditions and mixing within the reactor, which directly influence mass transfer and reaction rates. This highlights the need for advanced reactor designs that can maximize turbulence and mixing efficiency to complement catalyst-driven processes.

The hypothesis and objective of this study was to design and simulate a single-orifice oscillatory reactor and analyze the TKE, vorticity and dissipation factors within this reactor to increase mixing and also increase the produced biodiesel yield. In this study, the optimal reactor design dimensions were determined using numerical simulation and experimental analysis, and finally, solutions were presented to increase the efficiency and quality of biodiesel production.

## 2. Materials and methods

This section explains the numerical and experimental approaches employed to optimize a single-orifice oscillatory flow reactor for biodiesel production, with detailed descriptions provided in the following subsections. Fig 1 shows the flowchart of the reactor design process.

### 2.1. Oscillatory flow reactor

Oscillatory flow reactors are tubular reactors in which annular plate baffles are placed at equal intervals. These reactors can be used in both continuous and discontinuous processes. Among the advantages of this reactor is the increase in heat transfer, mass transfer, and mixing [18]. One of the types of oscillatory reactors is the single-orifice oscillatory flow reactor.

### 2.2. Geometry of the single-orifice oscillatory flow reactor

As a critical component in the biodiesel production process, namely the transesterification reaction, the design of a single orifice reactor includes several important aspects that can help to optimize performance and production efficiency. This

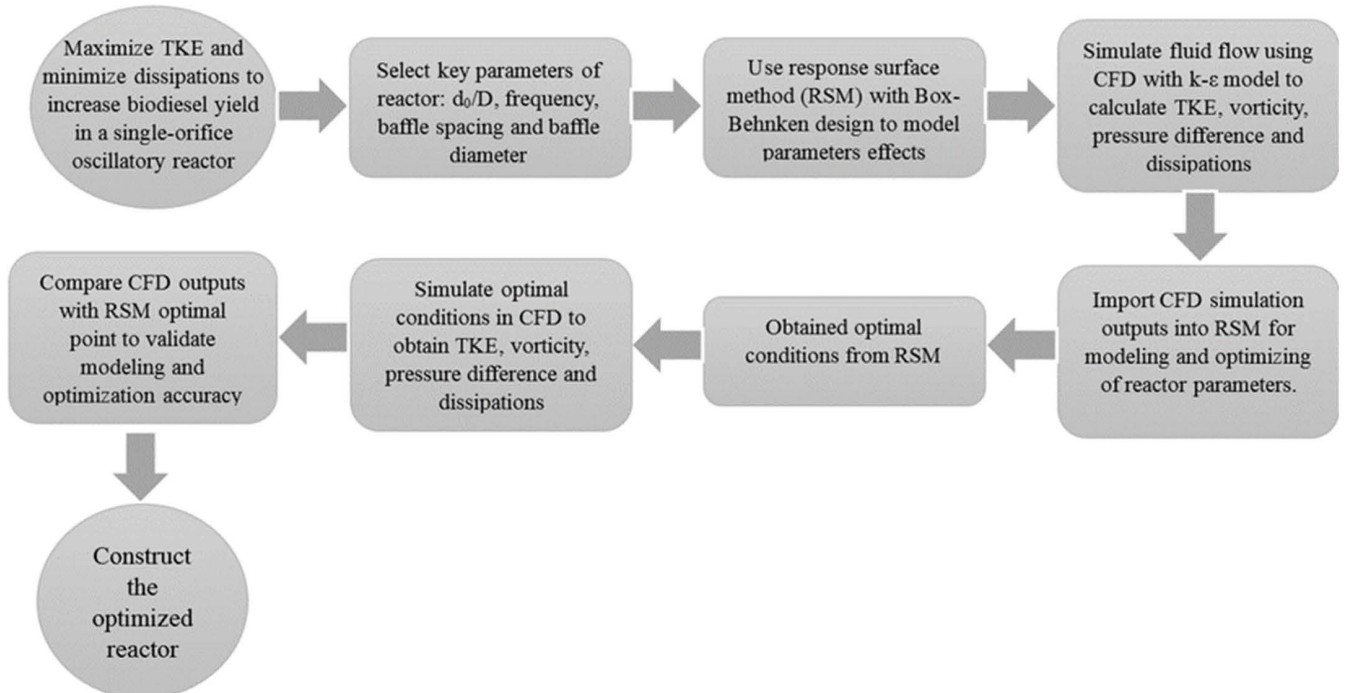

**Fig 1. Flowchart of parameter selection, simulation, and optimization for single-orifice oscillatory reactor.**

section will discuss the general structure, design parameters, and the effect of various variables on reactor efficiency. As a non-linear reactor, the single orifice reactor has advantages over traditional reactors due to its special mixing and flow behavior characteristics. The design of these reactors is mainly cylindrical or cubic, where complex slopes and walls can be designed to improve the flow and mixing of inputs. Single orifice oscillatory flow reactors are tubes in which baffles are placed at equal distances. A piston creates a reciprocating oscillatory movement. The oscillatory movement of the piston causes the intensification of the reactants. The behavior of the fluid inside the oscillatory reactors can be seen in Fig 2.

The reactor's dimensions and geometry greatly influence the hydrodynamic behavior and efficiency of the reactions. The selection of the diameter and height of the reactor, as well as the distance between the baffles and the diameter of the baffles, are among the important factors in the reactor design.

To determine the dimensions of the reactor, some geometric parameters, such as the distance between the baffles, the size of the baffles, and their dimensions, are needed. Also, three-dimensional dynamic parameters such as oscillatory Reynolds number, pure flow Reynolds number, and Strouhal number were investigated. Equation (1) was used to determine the diameter of the reactor.

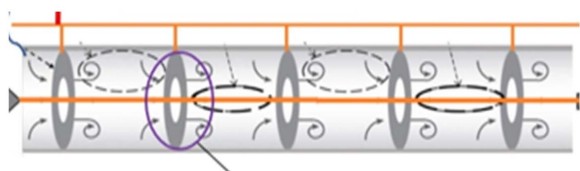

**Fig 2. Schematic of single-orifice oscillatory flow reactor design.**

The most important parameter in the flow is the fluctuating Reynolds number, which indicates the mixing intensity applied inside the pipe (Equation 1).

$$Re_o = \frac{2\pi f \rho D x_0}{\mu} \tag{1}$$

To create a turbulent flow inside the reactor, the Reynolds number between $10^4$ and $10^2$ was considered [19].

In this equation, D is the pipe diameter (m), ρ is the fluid density (kg/$m^3$, $X_0$ is the fluid amplitude (m), μ is the fluid viscosity (Pa.s), f is the oscillation frequency (Hz).

Strouhal number can be calculated from equation (2). Because the oscillatory flow occurs in this reactor without the intervention of the net flow, the Strouhal number is between 0.8 and 0.4 [20].

$$Str = \frac{D}{4\pi x_0} \tag{2}$$

In this equation, D is the diameter of the pipe (m), and $X_0$ is the amplitude of the oscillatory (m).

The Reynolds number of the additional net flow inside the pipe is also obtained with the help of equation (3).

$$Re_n = \frac{\rho u D}{\mu} \tag{3}$$

In this regard, D is the diameter of the pipe (m), μ is the dynamic viscosity of the fluid (Pa.s), ρ is the density of the fluid (kg/$m^3$) and u is the velocity of the fluid (m/s).

Equation (4) also shows the speed ratio [21].

$$\psi = \frac{Re_o}{Re_n} = \frac{2\pi x_0 f}{u} \tag{4}$$

In this equation, $X_0$ is the fluid amplitude (m), u is the fluid speed (m/s), and f is the oscillation frequency (Hz).

The distance ratio between baffles is also obtained from equation [11,13].

$$\frac{l}{D} = 0.5 - 1.5 \tag{5}$$

Also, the diameter of the successive convergence sections of the pipe is in the form of (6) [11,20].

$$d_0 = 0.4\, D \tag{6}$$

According to the above equation, the range obtained for the reactor design of the current research is shown in Table 1.

## 2.3. CFD simulation of the system

This research used the CFD method to perform single-phase simulation. CFD simulations were conducted to quantify TKE, vorticity and dissipations, enabling the evaluation of mixing efficiency within the single-orifice oscillatory reactor.

**Table 1. Range of reactor design parameters.**

| The ratio of baffle diameter to reactor diameter | (0.4-0.8) |
|---|---|
| Frequency (Hz) | (5-15) |
| Distance between baffles (mm) | (3-7) |
| The diameter of the baffles (mm) | (10-20) |

The simulation was performed in a time-dependent mode. Fluid flow was simulated under turbulent conditions and k-ε turbulent model. The mesh used in this simulation was normal. The input parameters include the ratio of baffle diameter to reactor diameter, the diameter of the baffles the distance between the baffles in millimeters, and the frequency in Hz. These input parameters have been used to investigate their effect on the output variables, which include TKE (maximum and average), turbulent energy dissipation rate, pressure difference, vorticity magnitude (maximum and average), and total viscous dissipation.

**2.3.1. Mesh independence and boundary conditions.** To ensure the accuracy and reliability of the CFD simulations, a mesh independence study was conducted. The computational domain was discretized using a structured mesh with tetrahedral elements, and three mesh resolutions were tested: coarse (50,000 elements), normal (150,000 elements), and fine (300,000 elements). The TKE and pressure difference were monitored as convergence criteria. The normal mesh (150,000 elements) showed a less than 2% deviation in TKE and pressure difference compared to the fine mesh, while the coarse mesh exhibited significant discrepancies (>10%). Thus, the normal mesh was selected to balance computational efficiency and accuracy.

Boundary conditions were carefully defined to reflect the physical behavior of the single-orifice oscillatory flow reactor. The reactor inlet was assigned a sinusoidal velocity profile to simulate the oscillatory flow, with the amplitude and frequency varying according to the input parameters (5–15 Hz). The outlet was set to a zero-gauge pressure condition to allow free outflow. No-slip boundary conditions were applied to the reactor walls and baffle surfaces, assuming zero velocity at these interfaces to account for viscous effects. The fluid properties, including density ($\rho = 880\,kg/m^3$) and viscosity ($\mu = 0.03\,Pa\cdot s$), were based on sunflower, consistent with the experimental conditions. The turbulence intensity at the inlet was set to 5%, a standard value for turbulent pipe flows, to initiate the k-ε model calculations.

**2.3.2. K-ε turbulent model.** Navier-Stokes equations using the Reynolds averaging method (RANS) are the governing equations of fluid flow used to describe fluid flow in this research. As mentioned earlier, CFD was used to perform this simulation. The k-ε model is one of the most common turbulent models in CFD, which is used to predict the behavior of turbulent flows. This model includes two equations, one for the TKE (k) and the other for the decay turbulent energy dissipation rate (ε). Due to its sensitivity to boundary conditions and input parameters, this model may sometimes require special settings. It is especially suitable for simulating turbulent flows with large changes [19,22]. But the main reason for choosing k- ε model for CFD simulation is its efficiency and reliability in simulating the intense turbulent flows of oscillatory reactor which are characterized by high Reynolds numbers and complex flow patterns. It excels at capturing TKE and its dissipation rate, crucial for enhancing mixing in biodiesel production. While models like k-ω SST offer precision near walls, k-ε strikes an effective balance between accuracy and computational simplicity, ideal for focus on bulk flow dynamics. Also, Large Eddy Simulation (LES), while offering higher accuracy for transient and small-scale turbulent structures, was deemed computationally prohibitive for the iterative optimization process involving multiple reactor configurations. Thus the k-ε making it ideal for this study's focus on optimizing reactor geometry and operating conditions.

$$\mu_t = \rho C_\mu \frac{k^2}{\epsilon} \tag{7}$$

$$\frac{\partial}{\partial_t}(\rho k) + \frac{\partial}{\partial x_i}(\rho k u_i) = \frac{\partial}{\partial x_j}\left(\left(\mu + \frac{\mu_t}{\sigma_k}\right)\frac{\partial_k}{\partial x_j} + G_k + G_b - \rho\epsilon - Y_M + S_k\right) \tag{8}$$

$$\frac{\partial}{\partial_t}(\rho\epsilon) + \frac{\partial}{\partial x_i}(\rho\epsilon u_i) = \frac{\partial}{\partial x_j}\left\{\left(\mu + \frac{\mu_t}{\sigma_\epsilon}\right)\frac{\partial}{\partial x_j}\right\} + C_{1\epsilon}\frac{\epsilon}{k}(G_k + G_{3\epsilon}G_b) - C_{2\epsilon}\rho\frac{\epsilon^2}{K} + S_\epsilon \tag{9}$$

Effective components in transfer equations:

$G_k$ :is the energy production due to the gradients of the average speed of the flow.

$G_b$: is the energy production due to buoyancy (it is zero for flows without gravity and heat transfer).

$Y_M$: Kinetic energy production due to flow compressibility effects (it is zero for incompressible flows).

$\sigma$, $\sigma_k$: Parantel numbers proportional to k and ε.

$S$, $S_k$ :spring terms definable by the operator.

### 2.3.3. Turbulent Kinetic energy (TKE).

TKE refers to a part of fluid kinetic energy that is caused by random and unstable movements of particles in a fluid flow. This energy refers to the distribution and changes of speed inside a flow compared to its average speed and generally indicates the intensity and strength of turbulence in the fluid. In other words, TKE affects the flow and behavior of the fluid with irregular and random movements. The TKE is one of the key parameters in process engineering and fluid dynamics because it directly affects the mass and heat transfer rates in various systems. In reactors, TKE improves mixing, especially in processes such as biodiesel production that require optimal mixing of raw materials. The turbulence kinetic energy is defined as equation (9).

$$\text{TKE} = \frac{1}{2}((\bar{u}\,')2 + (\bar{v}\,')2 + (\bar{w}\,')2)$$

(9)

In (9), $w'$, $v'$, $u'$ are the speed deviations in the three coordinate directions (x, y, z), respectively.

Finally, the kinetic energy of turbulence is considered a key parameter in the analysis and optimization of chemical reactors with high potential for biodiesel production. A detailed understanding of this energy's behavior and the factors affecting it can lead to the development of optimal and more sustainable processes in the production of biofuels.

### 2.3.4. Turbulent energy dissipation rate.

Turbulent energy dissipation rate is a measure to measure the amount of energy in turbulent flows due to the existence of fluctuations and unpredictable movements, continuous energy from large scales (such as the main flow) to small scales (such as very fine turbulence). It is transferred. This energy eventually turns into heat due to friction and destruction. Turbulent energy dissipation rate depends on various factors, including fluid characteristics, turbulent intensity, and environmental conditions.

$$\varepsilon = \frac{1}{2}v(\overline{\frac{\partial ui}{\partial xj}} + \overline{\frac{\partial uj}{\partial x}})^2$$

(10)

In this regard, ε is the turbulent energy dissipation rate (m²/ s³), v is the fluid dynamic viscosity (m²/s), and u is the velocity fluctuations.

### 2.3.5. Pressure difference.

The pressure difference in a single orifice oscillatory reactor depends on various factors and is usually explained based on the dynamic interactions between the liquid or gas flow and the reactor structure. In an oscillatory reactor, the fluid flow is usually oscillatory and changing. These fluctuations create zones with different pressures inside the reactor. Various factors, such as changes in speed, temperature, and fluid composition, can cause fluctuations in flow. In oscillations, gravitational and frictional forces between the fluid and the reactor walls can also lead to a pressure difference. When the fluid is moving, friction causes some of the kinetic energy to be converted into heat, which in turn can affect the pressure. If the fluctuations are periodic, this can lead to a pressure difference at the reactor outlet compared to the inlet. The pressure difference in an oscillatory reactor can be considered as a result of a combination of flow fluctuations, gravitational forces, friction, and reactor design. Numerical models and computer simulations can be used for more detailed analysis to predict flow behavior and pressure differences. Equation (11) shows the pressure difference.

$$\Delta P = P_2 - P_1$$

(11)

$P_2$ (Pa) is the highest reactor pressure, and $P_1$ (Pa) is the lowest reactor pressure.

**2.3.6. Viscous dissipation.** Viscous dissipation refers to the amount of energy converted into heat as a result of the fluid's internal friction during its movement. This phenomenon has a major effect on the performance of fluid systems, especially in turbulent and continuous flows. Viscous dissipation is usually represented by the symbol ($\Phi$) and is defined in the form of equation (12).

$$\Phi = 2\mu \left(\frac{\partial u}{\partial x}\right)^2 + 2\mu \left(\frac{\partial v}{\partial y}\right)^2 + 2\mu \left(\frac{\partial w}{\partial z}\right)^2 \tag{12}$$

In this regard, μ is the dynamic viscosity of the fluid, and u, v, and w are the components of the fluid velocity in three coordinate directions.

## 2.4. Modeling and optimization of simulation parameters

The aim of this research is optimizing the geometric dimensions of the oscillatory reactor and investigate the effect of different geometric variables on the output parameters of the simulation. For this purpose, the RSM has been used. Modeling and optimization using the RSM was performed to optimize the reactor design parameters, increase mixing and agitation to achieve the highest biodiesel production efficiency within the reactor. This method models a non-linear equation between input and output variables and facilitates optimization. By specifying the upper and lower limits for each variable and the number of variables, this method creates a test matrix and, based on it, determines the number of tests and the levels of each variable in each test. (RSM) is very efficient, especially when the number of variables is large, and to achieve optimal values, it takes advantage of solving the existing equation (13).

$$Y_i = \beta_0 + \sum \beta_i X_i + \sum \beta_{ij} X_i X_j + \sum \beta_{jj} X_i^2 + \tag{13}$$

In this regard, $\beta_0$, $\beta_i$, $\beta_{ij,}$ and $\beta_{jj}$: are constant coefficients, $X_i$ and $X_j$: are independent variables of the process, and ε is a random error. box Benken's design was used in this experiment. The selected levels for the independent variables of the experiment were chosen according to Table (2) and the influence of the independent parameters on the diameter of the ratio of baffle diameter to reactor diameter ($d_0/D$), frequency (Hz) (f), the distance between baffles (mm) (B), baffle diameter (mm) (C), were checked.

RSM used a Box-Behnken design to generate reactor configurations from independent variables (Table 2), and then each run extracted from RSM was simulated using CFD methods to get simulation outputs. These outputs were analyzed in RSM, identifying an optimal design. The optimum point was simulated with using CFD methods and the results were compared with RSM optimization and predictions to match the results and obtain the reactor dimensions.

## 2.5. Experimental test and validation

In this study, sunflower oil was used as the raw material. Potassium hydroxide tablets (KOH) 99.8% were used as the homogeneous catalyst and methanol 99.9% (manufactured by Merck, Germany) was used as the alcohol of the experiment. A mixture of methanol and KOH were used in the transesterification reaction as methoxide, which was conducted

**Table 2. Codded levels of independent variables in the RSM.**

| Independent variable | Codded level | | |
|---|---|---|---|
| | −1 | 0 | 1 |
| The ratio of baffle diameter to reactor diameter | 0.4 | 0.6 | 0.8 |
| Frequency (Hz) | 5 | 10 | 15 |
| Distance between baffles (mm) | 3 | 5 | 7 |
| The diameter of the baffles (mm) | 10 | 15 | 20 |

under ideal circumstances determined by the simulations. The molar ratio of alcohol to oil was maintained at 6:1, and the catalyst concentration was set at 1% w/w. To reproduce the settings examined in the simulations, oscillatory flow reactors (single orifice) with different geometries were constructed. Important factors, including oscillation frequency, spacing, and baffle diameter, were changed to match the simulated designs. Gas chromatography was used to measure the biodiesel yield and calculate the conversion rate. The link between TKE and vorticity with biodiesel yield was then examined by comparing the simulated findings with the experimental data. This method confirmed that the computer model accurately predicted the reactor's performance.

**2.5.1. Methyl esters (biodiesel) characterization.** A product sample (5 mL) was collected at the end of each experiment to determine the fatty acid methyl ester (FAME) content. Separation and detection of biodiesel products were performed by GC/MS (TSQ Quantum XLS Ultra) equipped with a 30 m long, 0.53 mm inner diameter Rt-U-BOND capillary column. Argon gas (99.99% pure) was used as the carrier gas. The temperature profile started at 40°C for 5 min and increased to 210°C at a heating rate of 6°C/min for 5 min. The total time of GC analysis was 60 min. The injector, ion source, and transfer line temperatures were maintained at 250°C, 180°C, and 180°C, respectively. The transfer line temperature was 180 °C. Finally, the FAME content of biodiesel was calculated using the equation.

$$\frac{(\text{FFA}_B - \text{FFA}_A) * 100}{\text{FFA}_B} \tag{14}$$

In this equation, $\text{FFA}_B$ is the free fatty acid content of the sample before treatment, and $\text{FFA}_A$ is the free fatty acid content of the sample after treatment.

## 3. Results and discussion

This study used an oscillatory flow reactor, which resulted in greater agitation and increased kinetic energy (TKE). This design's Important parameters include the baffle diameter ratio to reactor diameter ($d_0/D$), frequency (f), distance between baffles, and baffle diameter. The convergence diameter of the tube is in the range of (0.4–0.8) mm, the frequency is in the range of (5–15) Hz, the distance between the baffles is in the range of (3–7) mm, and the diameter of the baffles is in the range of (10–20) mm. Also, design expert software and response surface methods have been used to optimize the dimensions of the reactor. The simulation of fluid flow inside the reactor was also done using the CFD method.

### 3.1. Response Surface Method (RSM) analysis

The reactor dimensions were optimized using the response surface method in the design expert software. Independent variables of the experiment include the ratio of baffle diameter to reactor diameter, frequency, distance between baffles, and baffle diameter. Maximum TKE, average TKE, turbulent energy dissipation rate, pressure difference, vorticity magnitude, and viscous dissipation are also dependent variables of the experiment. Finally, the analyses were performed, and real and coded equations were extracted. Relations (15–20) show real relations.

$$
\begin{aligned}
\text{TKE max} = &-5.05104 + 19.97813 * d_0/D + 1.18337 * f - 0.10716 * B \\
&- 0.077320 * C - 0.26178 * d_0/D * f + 0.11356 * d_0/D * B - 0.15515 * d_0/D * C \\
&- 3.46750E - 003 * f * B - 0.012265 * f * C + 3.25750E - 003 * B * C - 16.35273 * d_0/D^2 \\
&- 3.99887E - 003 * f^2 - 2.33042E - 003 * B^2 + 4.30163E - 003 * C^2
\end{aligned} \tag{15}
$$

$$
\begin{aligned}
\text{TKE average} = &-7.73107 + 22.22719 * d_0/D + 1.04081 * f + 0.074229 * B \\
&+ 0.054035 * C - 0.12760 * d_0/D * f + 0.054187 * d_0/D * B - 0.20060 * d_0/D * C \\
&- 2.63000E - 003 * f * B - 0.010872 * f * C - 8.90000E - 004 * B * C - 17.09250 * d_0/D^2 \\
&- 4.91850E - 003 * f^2 - 7.00625E - 003 * B^2 + 2.00450E - 0903 * C^2
\end{aligned} \tag{16}
$$

$$\text{Turbulent energy dissipation} = -1016.82466 + 2549.86583 * d_0/D$$
$$+ 76.58008 * f + 4.06690 * B + 7.34328 * C - 50.66875 * d_0/D * f + 7.64375 * d_0/D * B$$
$$- 14.11250 * d_0/D * C - 0.46435 * f * B - 2.75464 * f * C - 0.041500 * B * C$$
$$- 1791.41250 * d_0/D^2 + 3.58013 * f^2 - 0.47903 * B^2 + 0.48315 * C^2 \tag{17}$$

$$\Delta P = -3940.30202 + 13616.51588 * d_0/D + 62.48167 * f - 860.75741 * B$$
$$+ 217.79075 * C - 1718.81448 * d_0/D * f + 256.73687 * d_0/D * B - 621.97750 * d_0/D * C$$
$$+ 92.59783 * f * B + 30.60303 * f * C + 0.15803 * B * C + 2175.53469 * d_0/D^2$$
$$+ 70.19749 * f^2 + 30.27597 * B^2 + 0.78996 * C^2 \tag{18}$$

$$\text{Vorticity average} = -0.19627 + 1.28931 * d_0/D + 0.33446 * f - 0.071702 * B$$
$$- 0.087973 * C - 0.32376 * d_0/D * f + 0.082925 * d_0/D * B + 0.071308 * d_0/D * C$$
$$- 3.52675E - 003 * f * B - 9.51900E - 003 * f * C + 2.58000E - 003 * B * C - 1.15647 * d_0/D^2$$
$$+ 9.10874E - 003 * f^2 - 5.36292E - 004 * B^2 + 2.56329E - 003 * C^2 \tag{19}$$

$$\text{Vorticity max} = +214.87607 - 80.82787 * d_0/D + 0.85382 * f - 32.52681 * B$$
$$- 12.43167 * -13.42625 * d_0/D * f + 20.92312 * d_0/D * B - 1.24825 * d_0/D * C$$
$$+ 0.035250 * f * B - 0.048610 * f * C - 0.082050 * B * C + 35.22687 * d_0/D^2$$
$$+ 0.83761 * f^2 + 1.87258 * B^2 + 0.47787 * C^2 \tag{20}$$

By removing the non-significant coefficients and dimensioning the remaining coefficients from the real relations, the coded relations of each dependent variable were obtained according to relations (21–26).

$$\text{TKE max} = +7.30 - 0.80 * A + 3.73 * B - 0.096 * C - 0.74 * D - 0.26 * A * B$$
$$+ 0.045 * A * C - \quad 0.16 * A * D - 0.035 * B * C - 0.31 * B * D + 0.033 * C * D$$
$$- 0.65 * A^2 - 0.100 * B^2 - 9.322E - 003 * C^2 + 0.11 * D^2 \tag{21}$$

$$\text{TKE average} = +6.59 - 0.46 * A + 3.45 * B - 5.942E - 003 * C - 0.60 * D$$
$$- 0.13 * A * B + 0.022 * A * C\ 0.20 * A * D - 0.026 * B * C - 0.27 * B * D - 8.900E - 003 * C * D$$
$$- 0.68 * A^2 - 0.12 * B^2 - 0.028 * C^2 + 0.050 * D^2 \tag{22}$$

$$\text{Turbulent energy dissipation} = +371.60 - 56.00 * A + 370.70 * B - 2.81 * C - 71.92 * D$$
$$- 50.67 * A * B + 3.06 * A * C - 14.11 * A * D - 4.64 * B * C - 68.87 * B * D$$
$$- 0.41 * C * D - 71.66 * A^2 + 89.50 * B^2 - 1.92 * C^2 + 12.08 * D^2 \tag{23}$$

$$\Delta P = +6646.80 - 1801.39 * A + 6785.89 * B + 1048.79 * C + 875.62 * D$$
$$- 1718.81 * A * B + 102.69 * A * C - 621.98 * A * D + 925.98 * B * C + 765.08 * B * D$$
$$+ 1.58 * C * D + 87.02 * A^2 + 1754.94 * B^2 + 121.10 * C^2 + 19.75 * D^2 \tag{24}$$

$$\text{Vorticity average} = +0.84 - 0.37 * A + 0.81 * B - 0.048 * C - 0.25 * D - 0.32 * A * B$$
$$+ 0.033 * A * C + 0.071 * A * D - 0.035 * B * C - 0.24 * B * D + 0.026 * C * D$$
$$- 0.046 * A^2 + 0.23 * B^2 - 2.145E - 003 * C^2 + 0.064 * D^2 \tag{25}$$

$$\text{Vorticity max} = +35.88 - 17.39 * A + 44.99 * B - 4.25 * C + 1.30 * D - 13.43 * A * B$$
$$+ 8.37 * A * C - 1.25 * A * D + 0.35 * B * C - 1.22 * B * D - 0.82 * C * D$$
$$+ 1.41 * A^2 + 20.94 * B^2 + 7.49 * C^2 + 11.95 * D^2 \tag{26}$$

In the above relations, it is (A: the ratio of baffle diameter to reactor diameter, B: frequency, C: distance between baffles, D: baffle diameter). According to the relations (15–26), it can be concluded that the frequency had the greatest impact on the dependent variables.

### 3.2. Effect of ratio of baffle diameter to reactor diameter ($d_0/D$), the distance between baffles on TKE (maximum and average)

The effect of ($d_0/D$) and the spacing between baffles on the TKE in an oscillatory flow reactor is significant. The graph in Fig 3a shows the trend of TKE max changes due to changes in ($d_0/D$) and the distance between the baffles. As it is clear

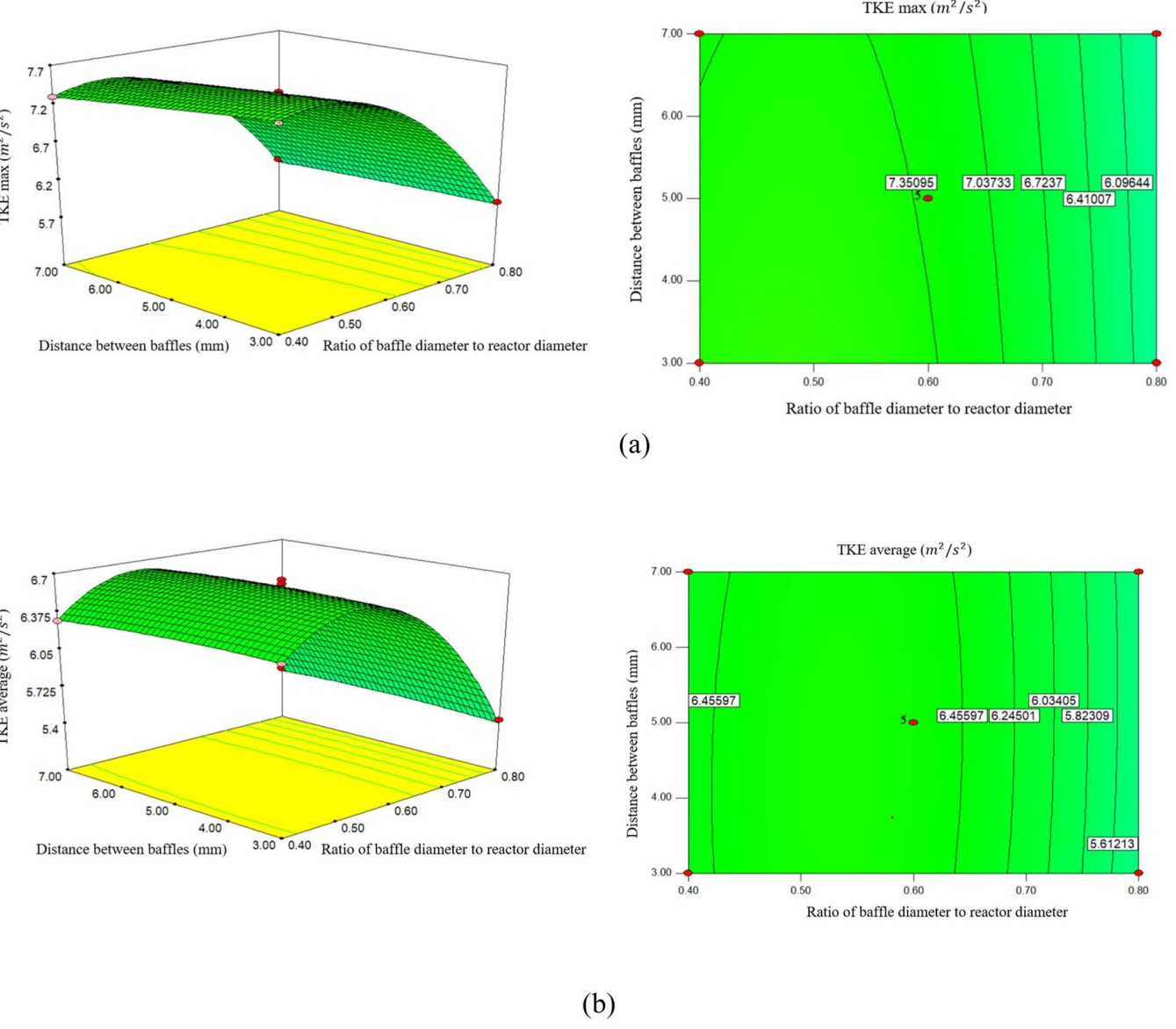

(a)

(b)

**Fig 3. The effect of the ratio of baffle diameter to reactor diameter and the distance between baffles on.** a) maximum Turbulent kinetic energy, **b)** Average of Turbulent kinetic energy.

from Fig 3a, with the increase of (d$_0$/D), the TKE max inside the reactor decreases with a gentle slope. Also, according to the diagram of changes in the diameter ratio, it is clear that the increase in diameter leads to a decrease in TKE max.

According to this diagram, it can be concluded that if (d$_0$/D) changes from 0.4 mm to 0.6 mm, TKE max, from 7.45192 ($m^2/s^2$) to 7.3662 ($m^2/s^2$) also changes, indicating a slight decrease in TKE max. If (d$_0$/D) increases and reaches 0.8 mm, TKE max reaches 5.84298 ($m^2/s^2$). As it is clear from this diagram, the increase of (d$_0$/D) has the opposite effect on the rise in TKE max inside the pipe. The effect of (d$_0$/D) on the TKE is significant and affects the flow characteristics and mixing efficiency [23,24]. While smaller (d$_0$/D) can increase max TKE and improve mixing, they may also lead to increased pressure drop and energy dissipation, requiring a balance between performance and efficiency in reactor design.

According to Fig 3b, with the increase of (d$_0$/D), the TKE average increases with a very low slope, and then with the increase of (d$_0$/D), the TKE average has a decreasing trend. So that (d$_0$/D) increases from 0.4 mm to 0.6 mm, the TKE average increases from 6.36408 ($m^2/s^2$) to 6.493 ($m^2/s^2$). When (d$_0$/D) increases from 0.6 to 0.8 mm, the TKE average decreases from 6.6493 ($m^2/s^2$) to 5.4449 ($m^2/s^2$). The (d$_0$/D) effect in alternating current reactors significantly affects the average TKE. Research shows that changes in tube diameter can alter flow patterns, mixing efficiency, and pressure drop, all of which are critical to optimizing reactor performance. The thickness-to-diameter ratio affects TKE. As this ratio increases, TKE tends to decrease, indicating that larger orifice diameters may reduce turbulence [25].

According to the study conducted by [26], a smaller (d$_0$/D) can increase the fluid velocity and increase TKE due to higher shear rate and turbulent generation. Conversely, larger diameters may reduce turbulence and result in lower TKE, affecting mixing efficiency and reaction rates.

It is generally expected that the flow conditions in the reactor will improve by increasing the ratio of baffle diameter to reactor diameter (d$_0$/D). Baffles, which are components that control fluid movement, can enhance mixing and minimize the occurrence of dead zones.

According to Fig 3a, TKE max decreases with a very gentle slope as the distance between the baffles increases. By increasing the distance between the baffles from 3 mm to 5 mm, the TKE max also increased from 7.38851 ($m^2/s^2$) to 7.3662 ($m^2/s^2$) and also by increasing the distance between the baffles from 5 mm to 7 mm TKE max also from 7.3662 ($m^2/s^2$) to 7.1959 ($m^2/s^2$) decreases. Closer cavities enhance the formation of vortices, which are critical for turbulent mixing and energy dissipation, leading to higher maximum TKE [6]. Conversely, while a narrower baffle spacing can increase TKE and heat transfer, it may increase energy consumption due to greater pressure drop. This trade-off requires careful design considerations to optimize performance without incurring excessive operating costs.

According to Fig 3b, as the distance between baffles increases, the TKE average increases with a very low slope, and as the distance between baffles increases, the TKE average decreases. When the distance between the baffles increases from 3 mm to 5 mm, TKE average also increases from 6.5661 ($m^2/s^2$) to 6.6493 ($m^2/s^2$), and when the distance between baffles increases from 5 mm to 7 mm, TKE average from 6.6493 ($m^2/s^2$) to 6.5542 ($m^2/s^2$) decreases. The spacing between baffles in an OFR significantly affects the system's average TKE. A closer baffle spacing increases the interaction between the oscillatory flow and the fluid, leading to more pronounced folding and stretching of the fluid, which causes turbulent mixing [27].

In a study conducted by [28] in an oscillatory flow reactor, the distance between baffles affects the residence time and flow pattern. Closer baffles can increase turbulence and TKE by promoting frequent interactions between the liquid and the baffles. Baffles that are too close may lead to flow stagnation and reduce overall efficiency.

The spacing between baffles also has a significant effect on TKE. Reducing the spacing between baffles results in more turbulence and better mixing. As the spacing between baffles increases, dead zones may increase, which can reduce TKE.

## 3.3. Effect of frequency and Baffle diameter on TKE (maximum and average)

The effect of frequency and baffle diameter on TKE is an important issue in the design of reactors and mixed vessels. These variables can significantly affect the flow pattern and TKE distribution. According to the analysis, the frequency

had the greatest effect on increasing TKE max. According to Fig 4a, the increase in frequency is a direct equation of the increase in TKE max. When the frequency increases from 5 Hz to 10 Hz, TKE max also increases from 3.47651 ($m^2/s^2$) to 7.3293 ($m^2/s^2$), and also when the frequency from 10 Hz increases to 15 Hz, TKE max also increases from 7.3293 ($m^2/s^2$) to 10.9267 ($m^2/s^2$). Research has shown that increasing the flow frequency can increase the maximum TKE. Higher frequencies increase vortex generation, increasing mixing and resulting in a more uniform flow pattern, similar to plug flow. Also, the oscillatory flow creates eddies that contribute to the overall mixing, thereby increasing the maximum TKE [28,29].

According to Fig 4b, the increase in the TKE average is directly correlated with the increase in frequency. When the frequency increases from 5 Hz to 10 Hz, the TKE average increases from 3.017 ($m^2/s^2$) to 6.6493 ($m^2/s^2$). By increasing the frequency from 10 Hz to 15 Hz, TKE ave increases from 6.6493 ($m^2/s^2$) to 9.134 ($m^2/s^2$). In oscillatory flow reactors, the oscillatory frequency can lead to changes in the shock structure and kinetic energy production. According to studies,

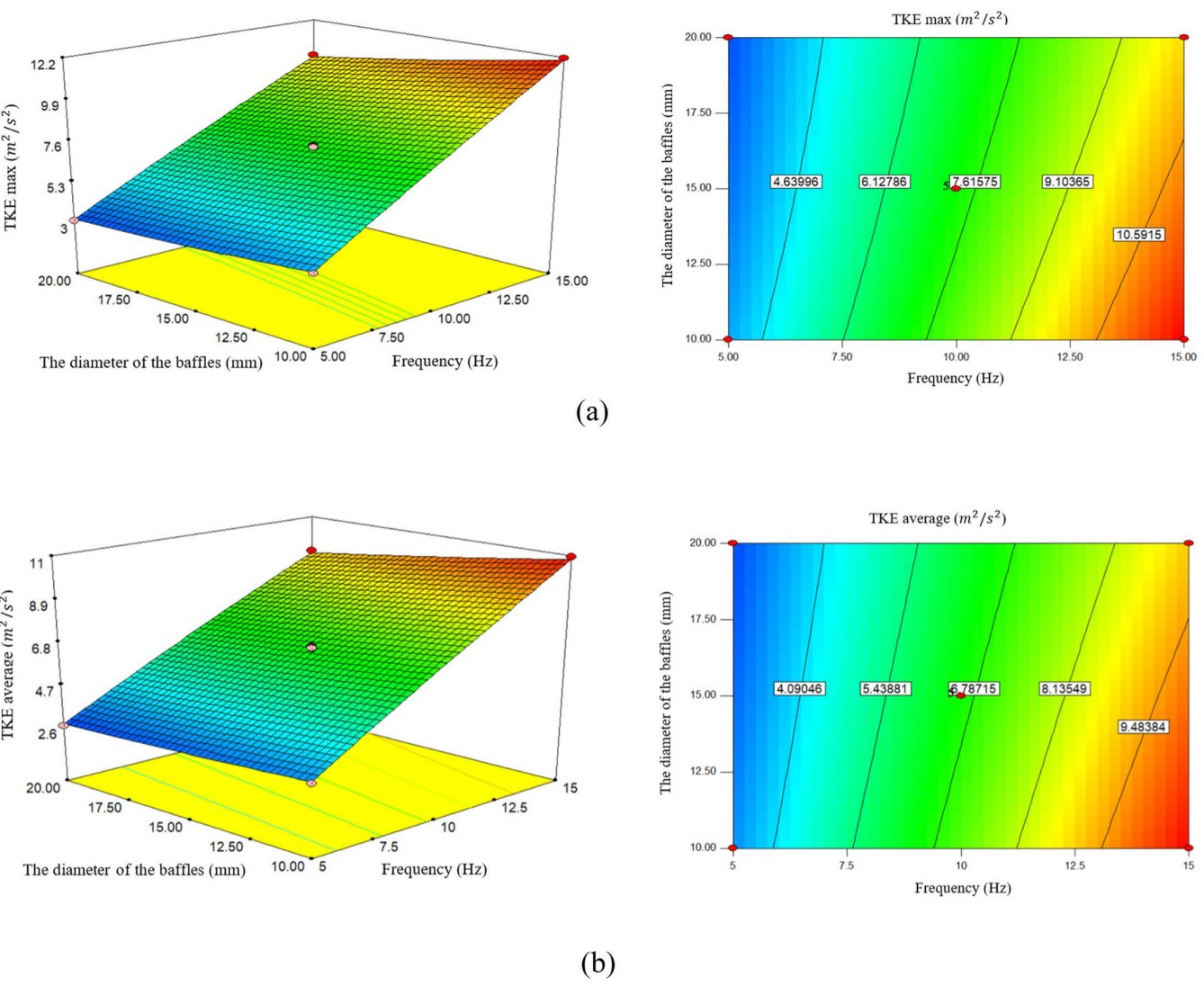

(a)

(b)

**Fig 4. The effect of frequency and the diameter of the baffles on.** a) maximum of Turbulent kinetic energy, **b)** Average of Turbulent kinetic energy.

increasing the oscillation frequency usually increases the intensity and turbulence of the flow. This leads to an increase in the average TKE.

The effect of frequency on average TKE in a single-orifice OFR is significant and affects mixing efficiency and reaction performance. Oscillatory motion increases fluid dynamics and results in improved mass and heat transfer, which is critical for optimizing reactor operation. Higher oscillation frequencies increase the mixing intensity, causing the formation of vortex rings in the baffle cavities, which increases the radial and axial mixing [30]. The design of OFRs can be optimized by adjusting the frequency to achieve desired TKE levels, increasing biodiesel production and other chemical processes [19,31].

According to the research done by [32], the reactor configuration significantly affects the turbulent and the reactor performance. Conversely, while higher frequencies can increase the disturbance energy, they may lead to instability in some systems, requiring frequency management. It shows accuracy to optimize the reactor performance without compromising the stability. Conversely, lower frequencies may result in insufficient mixing, leading to lower TKE and potentially slower reaction rates. This highlights the importance of frequency optimization for optimal reactor performance.

According to the research done by [33], increasing the oscillation frequency (3–14 Hz) increases energy dissipation and TKE, because higher frequencies promote more intense mixing and vortex formation. Selecting the optimal frequency to balance the energy input and mixing efficiency is essential. Also, a larger baffle diameter can cause significant flow disturbances and increase TKE.

According to Fig 3a, with the increase in the baffle diameter, TKE max has a decreasing trend. When the baffle diameter increases from 10 mm to 15 mm, TKE max increases from 8.14772 ($m^2/s^2$) to 7.3662 ($m^2/s^2$) and when the diameter of the baffle increases from 15 mm increases to 20 mm, TKE max from 7.3662 ($m^2/s^2$) to 67048.6 ($m^2/s^2$) decreases. Baffle diameter in oscillatory flow reactors significantly affects the system's maximum TKE. Research shows that changes in baffle geometry, including diameter, can alter flow patterns, energy dissipation, and mixing efficiency. The diameter of the baffles plays an important role in the TKE distribution. As the diameter of the baffles increases, they create significant disturbances in the flow, causing turbulence and increasing TKE. For example, configurations with optimal baffle dimensions have shown mixing performance that is almost twice as good as those without baffles [34]. While a larger baffle diameter may seem beneficial to reduce pressure drop, lower baffle diameters can compromise mixing efficiency and maximum TKE, highlighting the need for a balanced design approach. A larger baffle diameter can increase vortex formation and increase mixing and maximum TKE [35].

According to Fig 4b, increasing the baffle diameter decreases the TKE average with a gentle slope. Increasing the baffle diameter from 10 mm to 15 mm, TKE average from 7.2351 ($m^2/s^2$) to 6.493 ($m^2/s^2$) and also increasing the diameter from 15 mm to 20 mm, TKE average decreases from 6.6493 ($m^2/s^2$) to 6.041 ($m^2/s^2$). A larger baffle diameter can increase the mixing with a stronger oscillatory flow, resulting in a TKE average. Studies show that optimal baffle dimensions can lead to improved axial dispersion, which is essential for effective mixing in oscillatory baffle reactors [36]. Oscillation frequency and amplitude also interact with baffle diameter, affecting TKE and overall reactor efficiency [37,38].

The effect of baffle diameter and frequency on TKE is nonlinear. While TKE increases with increasing frequency, decreasing baffle diameter (increasing frequency) can also have a similar effect [39].

The increase in turbulent kinetic energy (TKE) with frequency, as shown in Fig 4a, is consistent with experimental data from a study conducted at frequencies up to 10 Hz. However, the results differ from those reported in [30], due to the smaller diameter of the reactor in this study, which limits vortex generation at higher frequencies. Regarding the effect of baffle diameter, the observed decrease in TKE contrasts with the findings of [33]. This discrepancy is because the spiral baffles used in this study do not use an aperture design, which could otherwise increase turbulence

The design parameters of an oscillatory flow reactor, such as ($d_0$/D), frequency, baffle spacing, and baffle diameter, significantly affect the TKE during biodiesel production. Understanding these effects is very important to optimizing the reactor.

The results obtained are consistent with the study conducted by [20,21]; therefore, decreasing ($d_0$/D) increases the TKE by 20% in the reactor. However, the latter shows a lower TKE. Compared to the studies conducted by [20,21], the sharp decrease in TKE is due to the difference in the amplitude of the oscillation. In this study, the baffle spacing shows a higher TKE than the study conducted by [24]. This indicates the important role of the baffle in creating turbulence.

While optimizing these parameters can increase TKE and reactor efficiency, it is necessary to consider trade-offs such as energy consumption and possible wear of reactor components. Balancing these factors is key to achieving optimal biodiesel production results.

Ultimately, careful investigation and experimental testing are necessary for each specific system to achieve the best settings and designs to increase TKE and improve mixing in the reactor [14].

### 3.4. Effect of ($d_0$/D), frequency, Baffle diameter, and distance between baffles on turbulent energy dissipation

The parameters ($d_0$/D), frequency, baffle diameter, and distance between baffles also affect the turbulent dissipation rate. Turbulent energy dissipation rate is one of the most important parameters in turbulent flow modeling. Since energy dissipation is never desirable, the diagram in Fig 5 shows the average turbulent energy dissipation due to changes in ($d_0$/D), frequency, baffle diameter, and distance between baffles. Fig 5a shows that with the increase of ($d_0$/D), the average turbulent energy dissipation increases with a slight slope at first and then decreases.

According to this diagram, when ($d_0$/D) increases from 0.4 mm to 0.6 mm, the energy dissipation rate is 355.93 ($m^2/s^3$) to 378.47 ($m^2/s^3$). Increases, if ($d_0$/D) increases to 0.8 mm, the energy dissipation rate decreases to 243.94 ($m^2/s^3$). Studies show that by increasing the diameter of the pipe, the surface cross-section is increased, and, as a result, the fluid flow rate is reduced. This means less friction with the pipe walls and less pressure drop. Smaller orifice diameters can increase energy dissipation by increasing turbulence, which improves mixing and mass transfer rates [40].

A study conducted by [26] showed that ($d_0$/D) affects the flow speed and turbulent intensity. A smaller diameter can increase velocity and result in higher turbulent energy, which is beneficial for mixing and reaction speed. Also, optimal convergence diameters can increase cavitation effects and further improve biodiesel production efficiency.

Increasing ($d_0$/D) can create more intense turbulence, which leads to an increased energy dissipation rate. However, if ($d_0$/D) is increased too much, it may lead to a decrease in mixing efficiency and the formation of dead zones [41].

According to Fig 5a, with the increase of the distance between the baffles, the energy dissipation rate increases with a very low slope at first and then decreases. By increasing the distance between the baffles from 3 mm to 5 mm, the energy dissipation rate increases from 372.48 ($m^2/s^3$) to 378.47 ($m^2/s^3$) and with increasing distance at 7 mm, the energy dissipation rate decreases to 87.366 ($m^2/s^3$). Baffles are important components in these systems that help to reduce uneven flow and create better mixing, the proper distance between them has a great impact on energy dissipation and overall system efficiency. Proper spacing between baffles helps improve fluid mixing. This mixing helps to reduce energy dissipation and improve system efficiency. Closer baffle spacing enhances turbulent mixing, enhances radial flow, and reduces energy dissipation due to increased interaction between the oscillatory flow and the baffles [27]. Studies show that the convergence diameter and the distance between the baffles affect the pressure drop and the energy dissipation rate, and optimization of the parameters minimizes the turbulent energy dissipation [35].

A study conducted by [42] found that closer spacing of baffles can increase turbulent energy dissipation and optimize the reaction environment for biodiesel production. On the contrary, optimizing these parameters can increase biodiesel production. Balancing energy input and operating costs is essential, as excessive turbulence may lead to increased wear of reactor components and higher operating costs.

As the spacing between baffles increases, the rate of energy dissipation decreases because less flow is directed towards the baffles and less turbulence is created. Conversely, a smaller spacing increases contact and turbulence, resulting in an increased rate of energy dissipation [43].

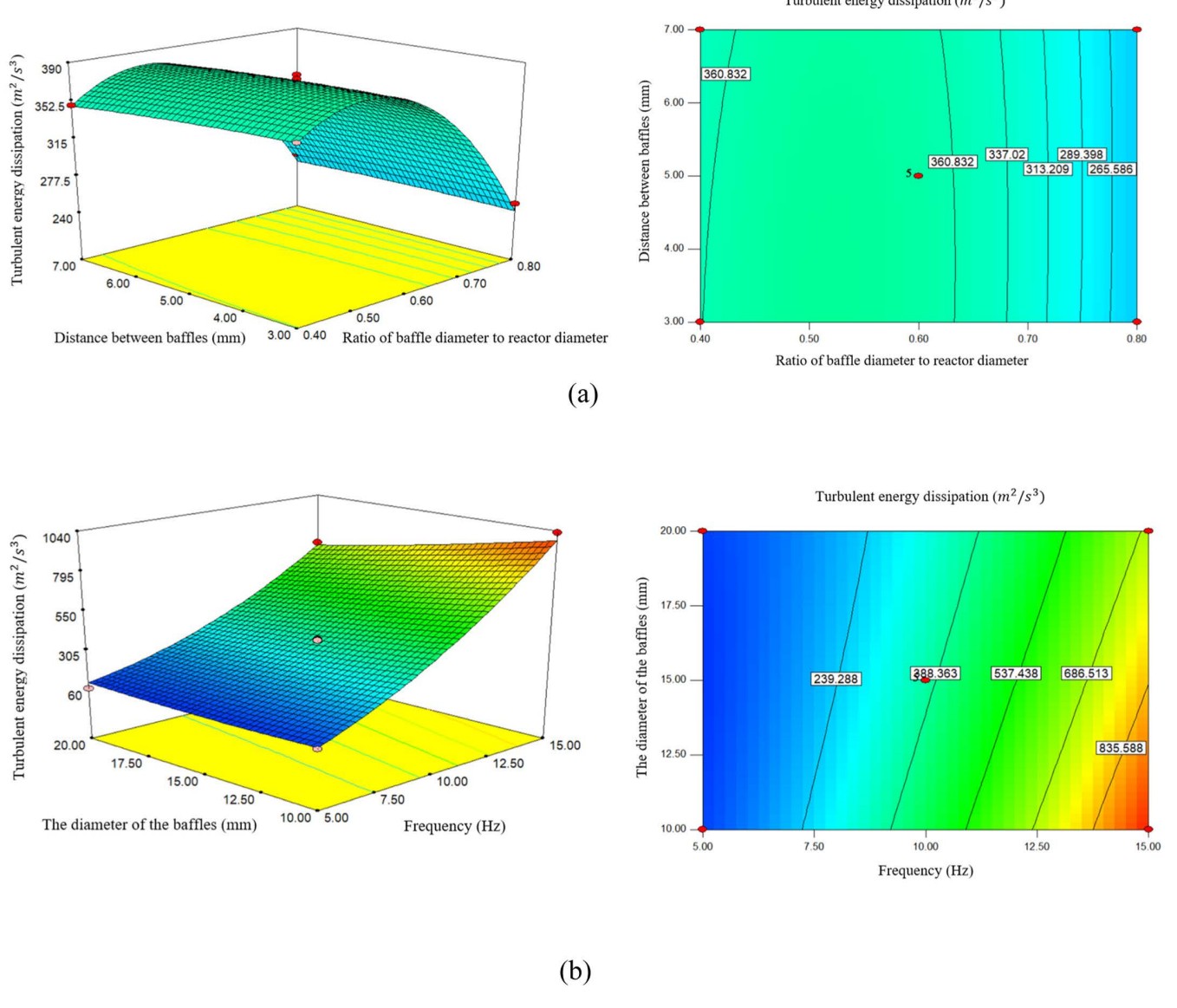

Fig 5. a) The effect of the ratio of baffle diameter to reactor diameter and distance between baffles on turbulent energy dissipation, b) The effect of frequency and the diameter of the baffles on turbulent energy dissipation.

According to Fig 5b, the energy dissipation rate increases with increasing frequency. By increasing the frequency from 5 Hz to 10 Hz, the energy dissipation rate from 90.39 ($m^2/s^3$) to 374.43 ($m^2/s^3$), and by increasing the frequency to 15 Hz, energy dissipation increases to 831.8 ($m^2/s^3$). Frequency is an important parameter in energy dissipation rate. In dynamic systems, an increase in the frequency of oscillations may increase energy dissipation and cause vibration and increased stresses in mechanical components. This phenomenon can lead to fatigue and even premature failure. Low frequencies can usually increase efficiency and reduce energy dissipation under certain conditions. In fluid transmission systems, the frequency of oscillations can affect the creation of vortices and non-linear flows. These fluctuations may increase pressure drop and, consequently, energy dissipation. Oscillatory flow reactors produce vortices that increase mixing and energy dissipation, with frequency playing an important role in vortex dynamics [28].

Research conducted by [28], frequency plays an important role in generating vorticity magnitude, which increases mixing and reduces energy dissipation. Higher frequencies can lead to more effective turbulent and improved mass transfer rates. Excessively high frequencies may lead to increased energy consumption without commensurate benefits in reaction rate [42].

As frequency increases (smaller spacing between baffles), the rate of turbulent energy dissipation increases because the heterogeneous flows collide more frequently, continuously converting energy to heat. Conversely, low-frequency results in areas of uniform flow and reduced turbulence, which can reduce the rate of energy dissipation [44].

According to Fig 5b, with the change in the diameter of the baffles, the rate of energy dissipation has a linear trend. By increasing the diameter of the baffles from 10 mm to 15 mm, the energy dissipation rate from 455.59 ($m^2/s^3$) to 47.378 ($m^2/s^3$), and by increasing the diameter to 20 mm, the energy dissipation rate decreases to 311.75 ($m^2/s^3$). According to studies, larger baffles may reduce pressure drop and improve heat transfer, if the diameter is too large, it may cause anomalies and disrupt fluid flow, leading to increased energy dissipation. The diameter of the baffles can also affect the flow patterns and fluid mixing. If the baffles are not designed and selected correctly, they may cause ineffective currents and waste energy. The energy dissipation rate is closely related to the baffle diameter. Larger diameters tend to increase turbulent intensity and increase mixing efficiency [45].

Also, increasing the ratio of baffle diameter to reactor diameter, increasing frequency, increasing baffle diameter, and decreasing baffle spacing will lead to more energy dissipation. In general, it can be concluded that the design parameters of an OFR significantly affect the turbulent energy dissipation during biodiesel production.

The energy dissipation rate obtained from the study at 15 Hz is 15% lower than that of [23]. This difference is due to the smoother baffle edges, which reduce viscous dissipation. The effect of ($d_0/D$) on the dissipation is consistent with the simulations performed in the study [23], and smaller apertures increase turbulence. The optimal baffle spacing for minimum dissipation in this study is different from the results obtained in the study [38]. Optimizing parameters has an effect on reactor performance and increasing biodiesel efficiency.

### 3.5. Effect of ($d_0/D$), frequency, distance between baffles, and Baffle diameter on the pressure difference

The design parameters of a single-orifice OFR significantly affect the pressure difference during biodiesel production. According to the diagram in Fig 6a, with the increase of ($d_0/D$), the pressure difference has a downward trend. As ($d_0/D$) increases from 0.4 mm to 0.6 mm, the pressure difference from 8535.22 (Pa) to 6664.97 (Pa), and with the increase of ($d_0/D$) to 0.8 mm, the pressure difference decreases to 4932.43 (Pa). According to the studies, the reduction of ($d_0/D$) leads to an increase in fluid flow speed. According to Bernoulli's law, as the flow rate increases, the pressure decreases. Therefore, when ($d_0/D$) decreases, the pressure difference between the inlet and outlet increases significantly. In the convergence regions, the flow may experience additional friction with the walls, which helps increase the pressure difference. As ($d_0/D$) decreases further, these effects may become more apparent.

A smaller ($d_0/D$) can increase pressure drop due to higher fluid velocity and turbulence and increased mixing and mass transfer. Experimental studies show that orifice diameter significantly affects oscillatory flow characteristics, with smaller orifices producing larger pressure differences due to increased resistance [28,46].

Research conducted by [42], smaller ($d_0/D$) can increase fluid velocity and lead to increased pressure drop due to higher flow resistance. Conversely, larger diameters may reduce pressure drop and can compromise mixing efficiency.

As ($d_0/D$) increases, the surface area in contact with the flow increases. This can lead to increased turbulence and better mixing, but it can also lead to an increase in the pressure difference in the system. At higher values of ($d_0/D$), the liquid flow may encounter more resistance, which increases the pressure difference. As this ratio increases, the pressure difference is expected to increase, especially if the baffles are designed to direct the flow into narrower areas [47].

According to Fig 6a, the pressure difference will increase with the increase of the distance between the baffles. By increasing the distance between the baffles from 3 mm to 5 mm, the pressure difference increases with a slight slope

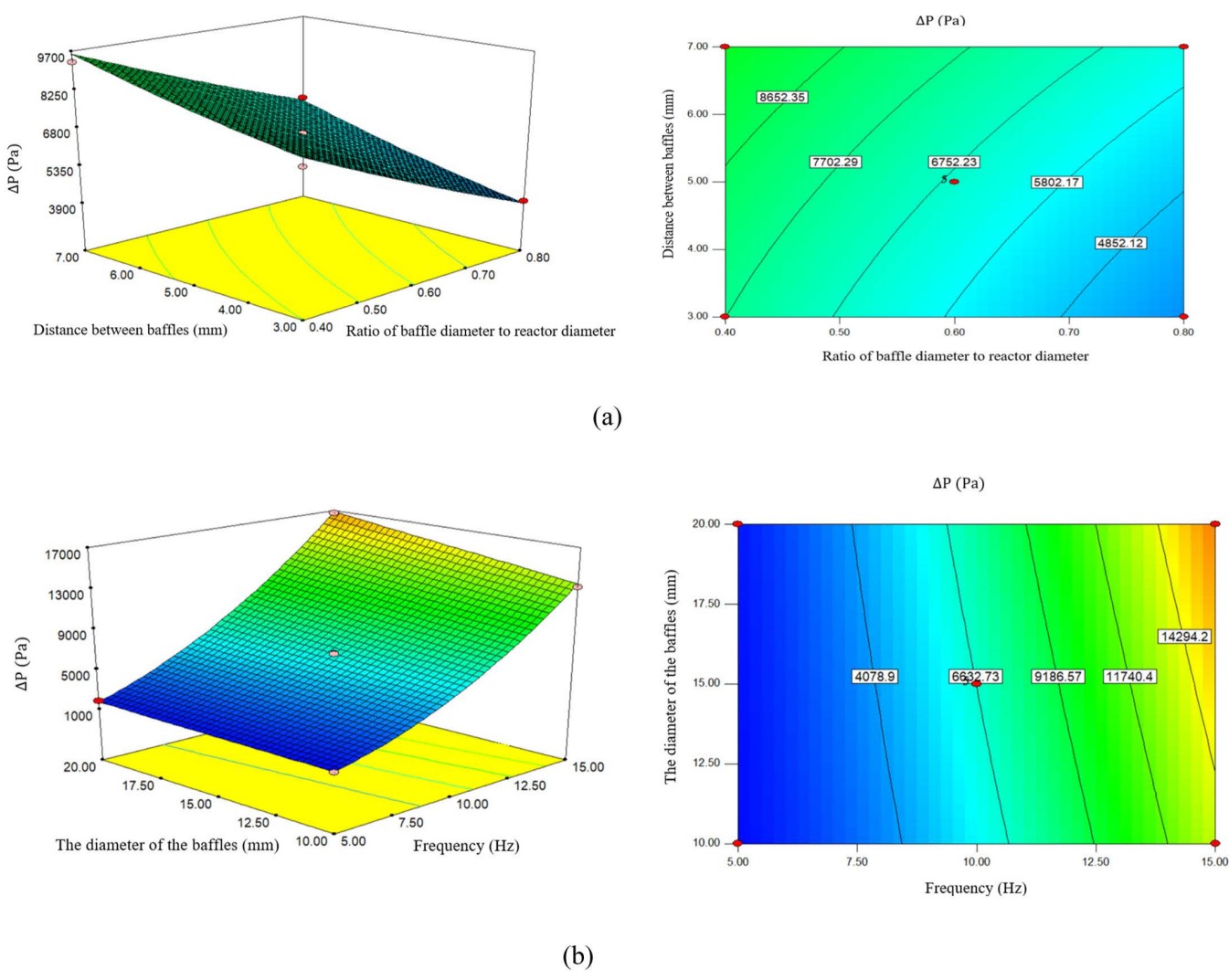

(a)

(b)

**Fig 6. a) The effect of the ratio of baffle diameter to reactor diameter and distance between baffles on pressure difference, b) The effect of frequency and the diameter of the baffles on pressure difference.**

from 12.5719 (Pa) to 97.6664 (Pa), and by increasing the distance to 7 mm, it increases to 69.7816 (Pa). According to research, the proper distance between the baffles can create a more homogeneous flow, which is the key to minimizing the pressure difference. Improper spacing can lead to vortices and low-pressure areas. Increasing the distance between the baffles may lead to a decrease in pressure drop in some cases, if the distance is too high, the system may not be able to use the mixing principles well, and as a result, the pressure difference will increase. The distance between the baffles changes the formation of vortices and the blockage of the flow, which directly affects the pressure drop. Closer baffles can increase mixing and may increase pressure drop due to restricted flow [35].

Reducing the spacing between baffles can lead to more turbulence and at the same time contribute to an increase in the pressure difference, as the fluid has to pass through more obstacles. While a larger spacing can facilitate flow and lead to a decrease in the pressure difference. To maintain turbulence and optimize mixing, the spacing between baffles must be carefully chosen to prevent excessive increases in the pressure difference [48].

According to Fig 6b, the increase in frequency is directly correlated with the increase in pressure difference. By increasing the frequency from 5 Hz to 10 Hz, the pressure difference increases from 1615.85 (Pa) to 6664.97 (Pa), and when the frequency reaches 15 Hz, the pressure difference also increases to 15187.6 (Pa).

According to the investigations, frequency changes can lead to variable pressure changes in fluid flow. In particular, with increasing frequency, pressure fluctuations usually increase, leading to intermittent pressure peaks and pressure drops. A higher frequency can lead to the creation of anomalies and non-linearity of the flow. These phenomena may lead to an increase in the pressure difference between different points in a system. Higher frequencies lead to increased pressure drop in oscillatory flow reactors. For example, at 50 Hz, the pressure drop can be two to three times higher than under constant flow conditions [49].

Higher oscillation frequencies improve mixing and increase mass transfer, which can lead to more uniform pressure distribution. According to the research conducted by [28], excessive frequency may lead to increased energy consumption without significant benefits.

Fig 6b shows that, as the diameter of the baffles increases, the pressure difference will also increase. By increasing the diameter from 10 mm to 15 mm, the pressure difference also increases from 5790.94 (Pa) to 6664.97 (Pa), and by increasing the baffle diameter to 20 mm, the pressure difference increases to 7542.17 (Pa). Does the change in the baffles' diameter affect the system's pressure drop? Baffles, as flow barriers, can cause changes in the velocity and pattern of fluid flow. When the diameter of the baffles becomes larger, the flow may be more homogeneous and face less pressure drop. Larger baffles can provide better mixing between different fluids. This mixing can reduce the pressure difference between different parts of the system. Larger baffle diameters can reduce pressure drop by allowing higher flow rates may also result in less effective mixing due to reduced shear. Conversely, smaller diameters can increase pressure drop and increase mixing through higher shear rates and turbulence [50]. While these parameters are critical to increasing the efficiency of biodiesel production, it is essential to consider the pressure drop and mixing effect to achieve optimal reactor design.

The pressure drop trend observed in this study is consistent with the experiments conducted by [25,40] where ($d_0$/D) decreases from 0.6 to 0.4, causing a 25% increase in ΔP. The absolute value of ΔP in this study is about 15% lower than theirs, which is due to the difference in fluid viscosity. The frequency-dependent increase in Fig 6b is consistent with the study conducted by [41]. The effect of baffle spacing on ΔP is consistent with the findings of [33]. The lowest value of ΔP in their study is reported to be 5 mm, while the value in Fig 6a increases.

Increasing the ratio of baffle diameter to reactor diameter results in a higher pressure difference. Additionally, increasing the frequency results in a higher pressure difference, while decreasing the frequency results in a lower pressure difference. Decreasing the distance between the baffles increases the pressure difference while increasing the distance decreases it. As the diameter of the baffles increases, the pressure difference also increases, especially when the baffles are designed to create more resistance to the flow [51].

### 3.6. Effect of ($d_0$/D) and distance between baffles on vorticity (max, average)

The mentioned parameters can greatly increase or decrease vorticity magnitude. According to Fig 7a, the average vorticity magnitude goes down with the increase of ($d_0$/D). By increasing ($d_0$/D) from 0.4 mm to 0.6 mm, the flow from 1.1630 (1/s) to 0.8445 (1/s), and by increasing the diameter to 0.8 mm to 0.4223 (1/s) decreases. Orifice geometry results in a two-fold increase in fluid circulation compared to pipe geometry at the same length-to-diameter ratio (L/D) [52].

According to Fig 7b, the maximum vorticity goes down with the increase of ($d_0$/D). Increasing ($d_0$/D) from 0.4 mm to 0.8 mm, the maximum vorticity decreases from 54.67 (1/s) to 19.90 (1/s). The increase of ($d_0$/D) has an inverse trend on the increase of maximum vorticity. Increasing orifice diameter reduces the range of oscillation frequencies, leading to a more stable flow regime and higher potential maximum vorticity [53].

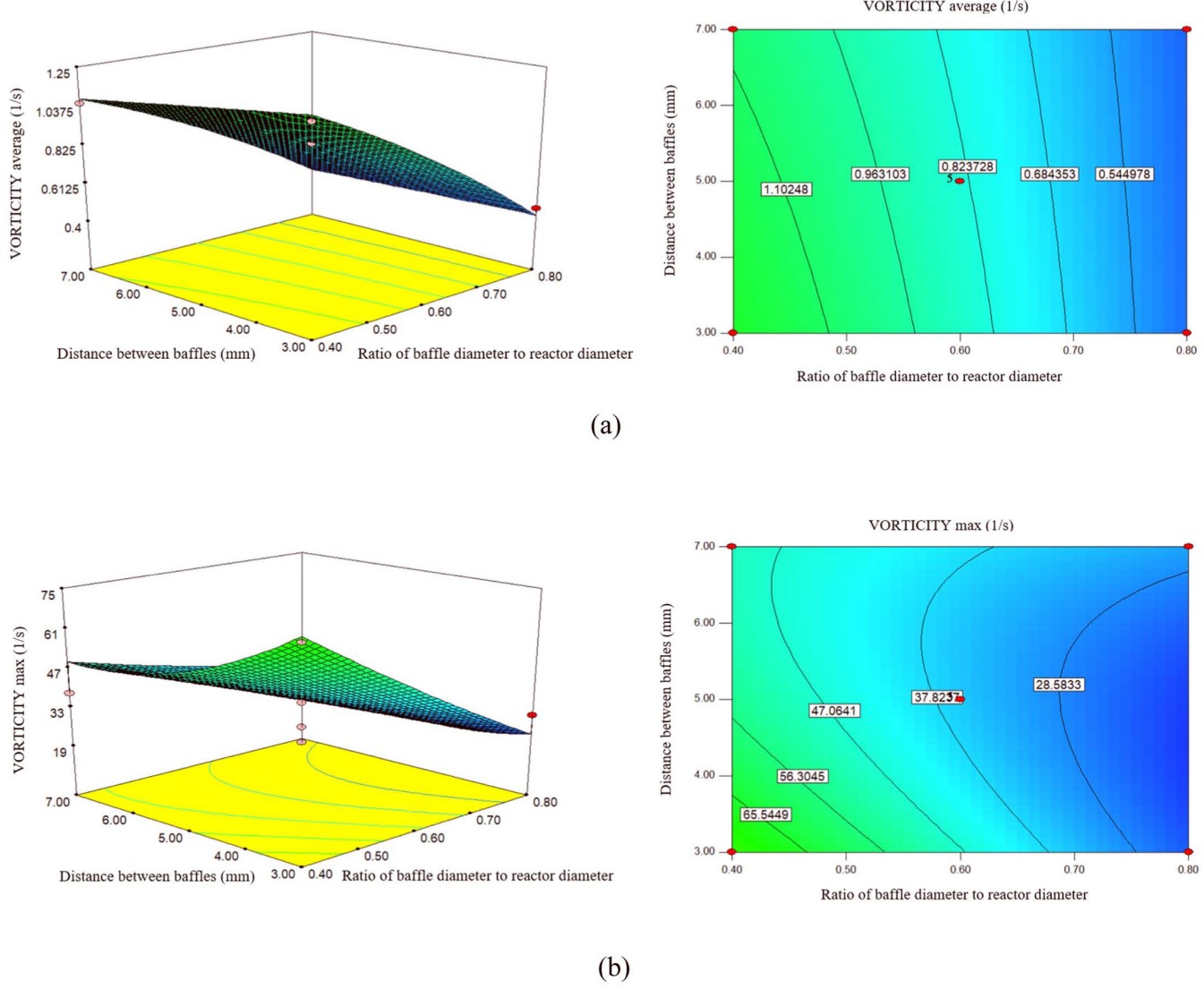

**Fig 7. The effect of the ratio of baffle diameter to reactor diameter and the distance between baffles on. a)** Average of Vorticity, b) maximum of Vorticity.

Research conducted by [28] showed that smaller ($d_0/D$) increases the fluid acceleration, which leads to increased vorticity magnitude generation as the fluid interacts with the baffles. This design can improve the mixing efficiency, which is important for biodiesel reactions requiring uniform conditions.

As ($d_0/D$) increases, more of the fluid surface is actually in contact with the baffles, which can increase turbulence in the fluid. This increased turbulence leads to an increase in the amount of vorticity. Increasing ($d_0/D$) usually results in an increase in the average and maximum vorticity, as the baffles convert the kinetic energy of the fluid into rotational energy, creating a form of instability and mixing. However, at very high values, the optimal vorticity may be reduced due to dead zones and flow constrictions [54].

Also, according to the diagram in Fig 7a, the average vorticity magnitude goes down with a very gentle slope with the increase in the distance between the baffles. By increasing the distance between the baffles from 3 mm to 5 mm,

the average vorticity magnitude from 0.8845 (1/s) to 0.8445 (1/s), by increasing the distance to 7 mm to 0.7890 (1/s). Decreased closer inter-baffle spacing creates more turbulent flow patterns, leading to less organized eddy flow, which can reduce mixing effectiveness [55]. On the contrary, the optimal distance allows better organization of vortices, increased mixing, and reduced dead zones in the reactor [50].

In Fig 7b, with the change in the distance between the baffles, the maximum vorticity has a downward trend with a very low slope. By increasing the distance between the baffles from 3 mm to 7 mm, the maximum vorticity decreases from 62.47 (1/s) to 12.39 (1/s). Closer baffles increase the interaction of the oscillatory flow with the fluid, leading to increased vortex formation. This is due to the radial flow induced by the oscillation, which is more pronounced when the baffles are closely spaced [27].

As the spacing between baffles decreases, more turbulent and unstable regions are formed, and this increased turbulence can lead to an increase in the amount of vorticity. Similarly, the average and maximum vorticity will also increase.

Conversely, a larger spacing between baffles can lead to a gradual decrease in flow and thus a decrease in vorticity, as the fluid passes through the baffles more easily and is less subject to rotation and oscillation.

### 3.7. Effect of frequency and Baffle diameter on vorticity (maximum, average)

According to Fig 8a, the average vorticity also increases as the frequency increases. By increasing the frequency from 5 Hz to 10 Hz, the average vorticity also increases from 0.2568 (1/s) to 0.8445 (1/s), and by increasing the frequency to 15 Hz to 1.8764 (1/s) increases. Higher oscillation frequencies increase vorticity generation, leading to increased turbulence and mixing in the reactor [32]. As the frequency increases, the flow exhibits an "overspeed" phenomenon, where the flow speed temporarily exceeds expected values and affects the vorticity levels [56].

By changing the frequency from 5 Hz to 15 Hz, the maximum vorticity increases from 11.83 (1/s) to 101.80 (1/s), which shows the direct effect of frequency on the maximum vorticity. Higher oscillation frequencies lead to increased vortex generation due to increased shear forces in the orifice [57].

A study conducted by [27] found that higher oscillation frequencies increase more intense fluid motion and form more vorticity magnitude. Increasing frequency can increase the mass transfer rate, which is beneficial for biodiesel production processes.

As the frequency increases, more turbulent and unstable regions are created in the flow. The fluid is forced to pass through narrow regions due to more obstacles, which leads to increased rotation and unstable flows. This usually results in increased vorticity (average and maximum). A larger distance between the baffles allows the flow to pass past the baffles more easily and creates fewer turbulent regions. As a result, vorticity (average and maximum) is reduced. Higher frequency usually contributes to increased vorticity, as it may create more turbulence and make the flow more turbulent [58].

According to Fig 8a, increasing the diameter of the baffles leads to a decrease in the average vorticity. By increasing the diameter from 10 mm to 15 mm, the average vorticity also decreases from 1.1559 (1/s) to 0.8445 (1/s), and by increasing the diameter to 20 mm to 0.6501 (1/s). A well-designed baffle can create a more uniform flow field, increase mixing, and reduce stagnant zones in the reactor [59]. Different baffle diameters can lead to different vortex structures due to blockage and flow reversal, which is essential for mixing efficiency [35].

According to Fig 8b, by changing the diameter of the baffles, the maximum vorticity magnitude will also increase. Increasing the diameter from 10 mm to 20 mm increases the maximum vorticity from 46.53 (1/s) to 49.12 (1/s). Eddy flow is a measure of rotation and instability of fluid flow and can help better understand flow patterns and energy transfer in fluid systems. A larger baffle diameter can increase the formation of vortices, which increases the mixing efficiency and may lead to a higher pressure drop [35].

A study conducted [60] in a tubular reactor with different transverse baffles revealed that a larger baffle diameter can create a significant flow blockage that leads to the formation of vorticity magnitude and increased mixing.

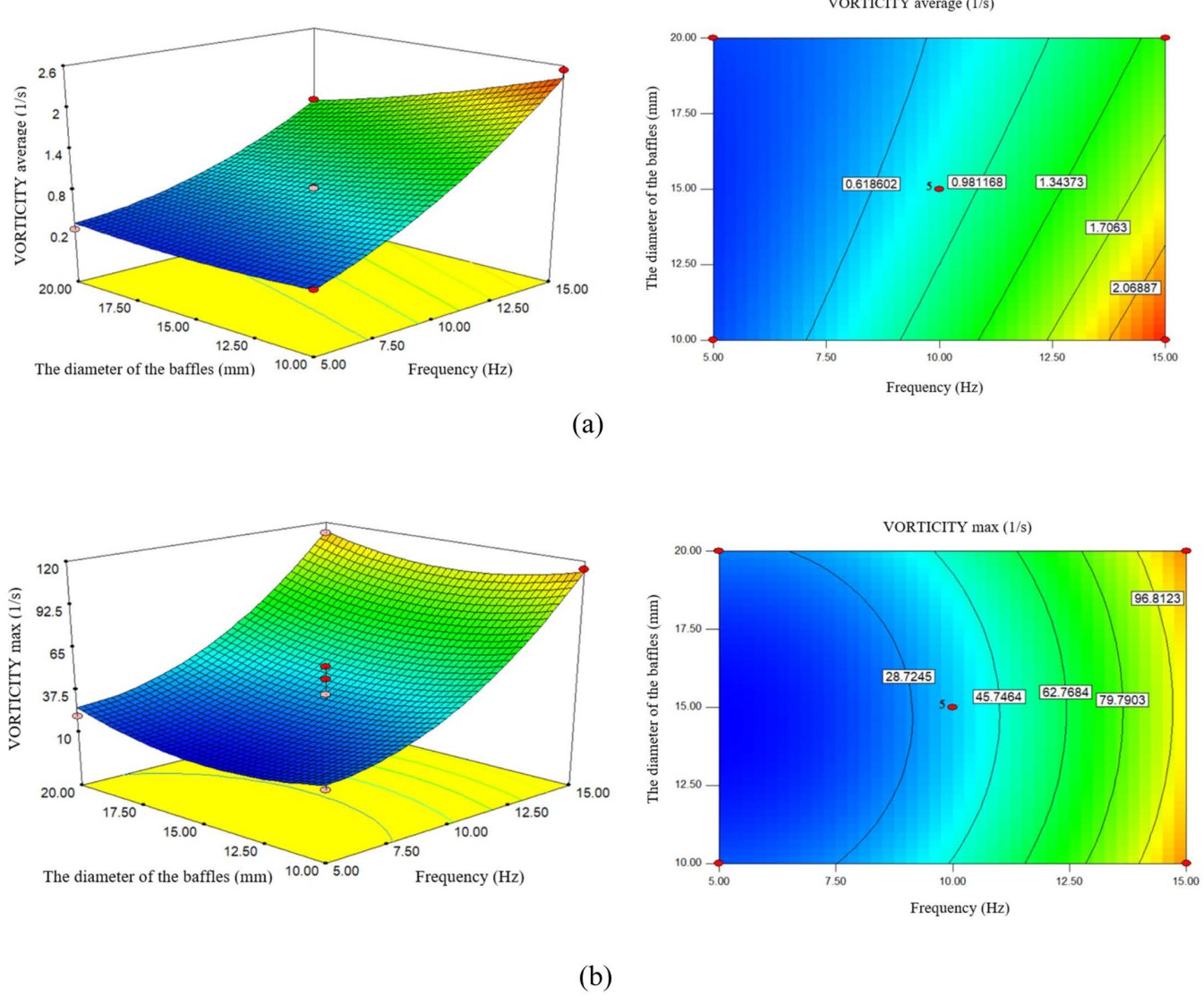

**Fig 8. The effect of frequency and the diameter of the baffles on a) Average of Vorticity, b) maximum of Vorticity.**

The geometry of the baffles also plays a role in determining the shear strain rate and energy dissipation, which are critical for optimizing biodiesel production [35].

Increasing the frequency can lead to more turbulence and increased vorticity (average and maximum). Also, the optimal frequency can maximize turbulence and mixing without creating dead zones. Increasing the diameter of the baffles helps increase vorticity, as it directs the flow towards turbulent and unstable areas. A properly designed baffle can optimize flow and mixing.

While these design parameters significantly increase mixing and vorticity magnitude. It is necessary to increase the efficiency of biodiesel production by optimizing the parameters.

The vortex flow in Figs 7 and 8 agrees with the result obtained in [22] that vortex formation increases with increasing baffle. The larger vortex size in this study than in them is due to the fluid viscosity. In contrast, the inverse relationship between $(d_0/D)$ and the vortex is not consistent with the findings of [40] and indicates the effect of the orifice on the flow.

## 3.8. Process optimization

The optimization of the single-orifice oscillatory flow reactor was performed using RSM with a Box-Behnken design to determine the optimal values for the independent variables: frequency (5–15 Hz), baffle diameter ratio ($d_0$/D, 0.4–0.8 mm), baffle spacing (3–7 mm), and baffle diameter (10–20 mm). The objective was to maximize TKE (maximum and average) and vorticity magnitude (maximum and average) while minimizing turbulent energy dissipation, pressure difference, and viscous dissipation to achieve the highest biodiesel yield. The RSM analysis generated a quadratic model (Equations (15–26) that correlated the input parameters with the output variables, and the optimal conditions were identified by solving the model to maximize TKE and biodiesel yield while keeping energy dissipation and pressure difference within acceptable limits.

This optimization showed that in the condition that ($d_0$/D), the distance between the baffles and the diameter of the baffles are respectively (3, 10, 0.4) mm and also the frequency is equal to (12.12) Hz, the maximum TKE and its average are respectively equal to 10.12 and 8.42 ($m^2/s^2$) and also the turbulent energy dissipation rate, pressure difference, and vorticity magnitude and its average are respectively equal to 655.06 ($m^2/s^3$), 9435.45 (Pa), 112.23 and 2.29 (1/s). This confirms that oscillatory flow reactors reduce energy consumption by 20% compared to traditional methods, aligning with sustainable production goals [61].

The proposed point and optimal conditions were finally simulated. They showed that in these conditions, the turbulent kinetic energy is 7.56 ($m^2/s^2$), the average turbulent energy is 6.36 ($m^2/s^2$), turbulent energy dissipation is 359.82 ($m^2/s^3$), the pressure difference is 7545.02 (Pa), and the average vorticity is 1.21 (1/s). Fig 9. The optimal conditions outperformed other tested ranges by maximizing TKE and vorticity and minimizing energy dissipations and pressure difference, leading

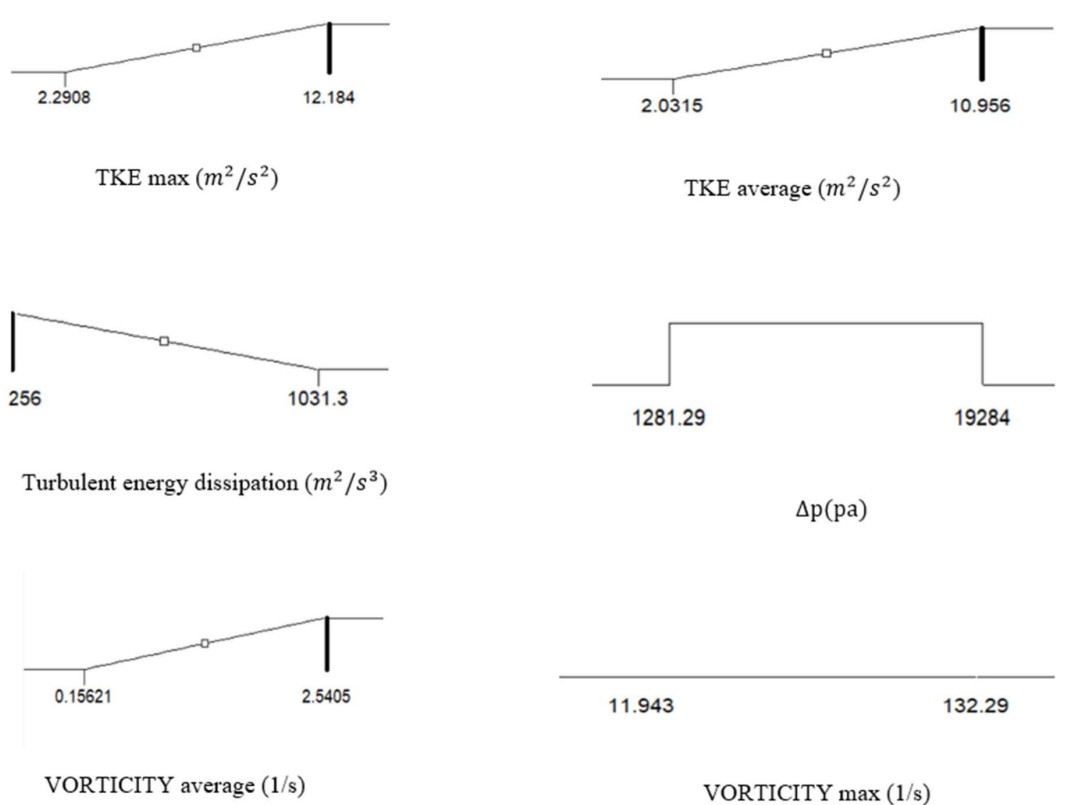

**Fig 9. Optimal reactor conditions determined by RSM analysis.**

to the highest biodiesel yield of 83% as validated experimentally. Compared to the study by [10], which reported an 89% biodiesel conversion using a single-orifice oscillatory flow reactor, present design achieves a slightly lower yield but with significantly reduced energy dissipation, representing a 20% improvement in energy efficiency. Additionally, the results outperform the smooth periodic constriction (SPC) reactor reported by [11],which achieved a 74.5% yield, demonstrating the superiority of our single-orifice design in enhancing turbulence and mixing efficiency. When compared to multi-orifice reactors [5], which achieved an 88% yield but required higher operating costs due to complex geometries, this simpler design balances performance and cost-effectiveness, making it more viable for industrial applications.

After obtaining the optimal point of this experiment by design expert software and the response surface method, the single orifice oscillatory flow reactor under these optimal conditions was carried out by (CFD) method. Fig 10 shows the contour of changes in speed, pressure, TKE, and turbulent energy dissipation inside this reactor under optimal conditions.

Fig 10a shows the fluid velocity in the single orifice oscillatory flow reactor. Here, the inlet of the reactor is located in the upper part of the reactor and the outlet is located in the lower part. According to this figure, the fluid velocity is slow around the baffles until near the center of the baffle. Also, according to the color scale of this diagram, it is clear that the velocity near the reactor walls (dark blue areas) is small. This low speed is because no-slip reactor walls are assumed in the simulation. As soon as the fluid enters the opening of the baffles, the speed starts to increase, and the fluid flow speed is the highest in the center of the baffles, which is equal to 0.0057 m/s. In fact, due to the decrease in the diameter of the pipe and also the pressure drops, the speed increases. Then, after the fluid passes through the opening of the baffles, the diameter increases again in this area, and the fluid speed decreases. As is clear in the figure, the fluid velocity in the center of the reactor is increased compared to the walls.

Fig 10b also shows the pressure distribution from the inlet to the reactor outlet. As it is clear from this figure, the maximum pressure occurs at the reactor outlet, which is equal to −30.30 (Pa). The lowest pressure occurs at the reactor inlet's beginning, equal to 7045.02 (Pa). As mentioned earlier, the pressure difference tends to decrease with the increase in the diameter of the tube convergence. Reducing the diameter at this point will lead to an increase in speed and a decrease in pressure. Also, the proper distance between the baffles minimizes the pressure difference.

Fig 10c also shows the turbulent energy dissipation rate. This parameter indicates the amount of energy lost as heat or other useless energy due to flow disturbances. As shown in the figures, the highest turbulent energy dissipation rate occurs at the end of the reactor, which is equal to 475.560 ($m^2/s^3$). This large amount of energy dissipation shows that many vortices are formed in this area, and a large part of fluid energy is wasted due to turbulence.

Fig 10d is related to the changes in the turbulent kinetic energy inside the single orifice oscillatory flow reactor. By examining the diagram, it can be concluded that the maximum amount of kinetic energy of turbulent is equal to 7.56 ($m^2/s^2$), which occurs in the end parts of the reactor. As can be seen, the most kinetic energy of disturbance occurs in the opening in the baffles. An increase in the kinetic energy of turbulence results in better mixing, as well as an increase in heat and mass transfer.

Finally, by analyzing these four figures, it can be concluded that by reducing the diameter in the center of the pipe, this reduction in diameter will increase the fluid speed, as well as increase the kinetic energy of the turbulent as well as pressure drop.

**3.8.1. Trade-offs between turbulence metrics and biodiesel yield.** The optimization process involved trade-offs between maximizing TKE and vorticity to enhance mixing and minimizing energy dissipation and pressure difference to ensure energy efficiency. Higher frequencies (15 Hz) increased TKE and vorticity but led to excessive energy dissipation (831.8 m²/s³), which could reduce the economic viability of the process. Similarly, a smaller $d_0/D$ (0.4 mm) enhanced turbulence but increased the pressure difference (7545.02 Pa vs. 4932.43 Pa at $d_0/D = 0.8$), requiring more robust reactor materials. Closer baffle spacings (3 mm) improved vorticity but increased energy dissipation and pressure drop, limiting their benefit to yield. Experimental validation confirmed a strong correlation between TKE ($r^2 = 0.972$) and vorticity ($r^2 = 0.882$) with biodiesel yield (Fig 11), supporting the selection of these parameters. The chosen conditions outperformed

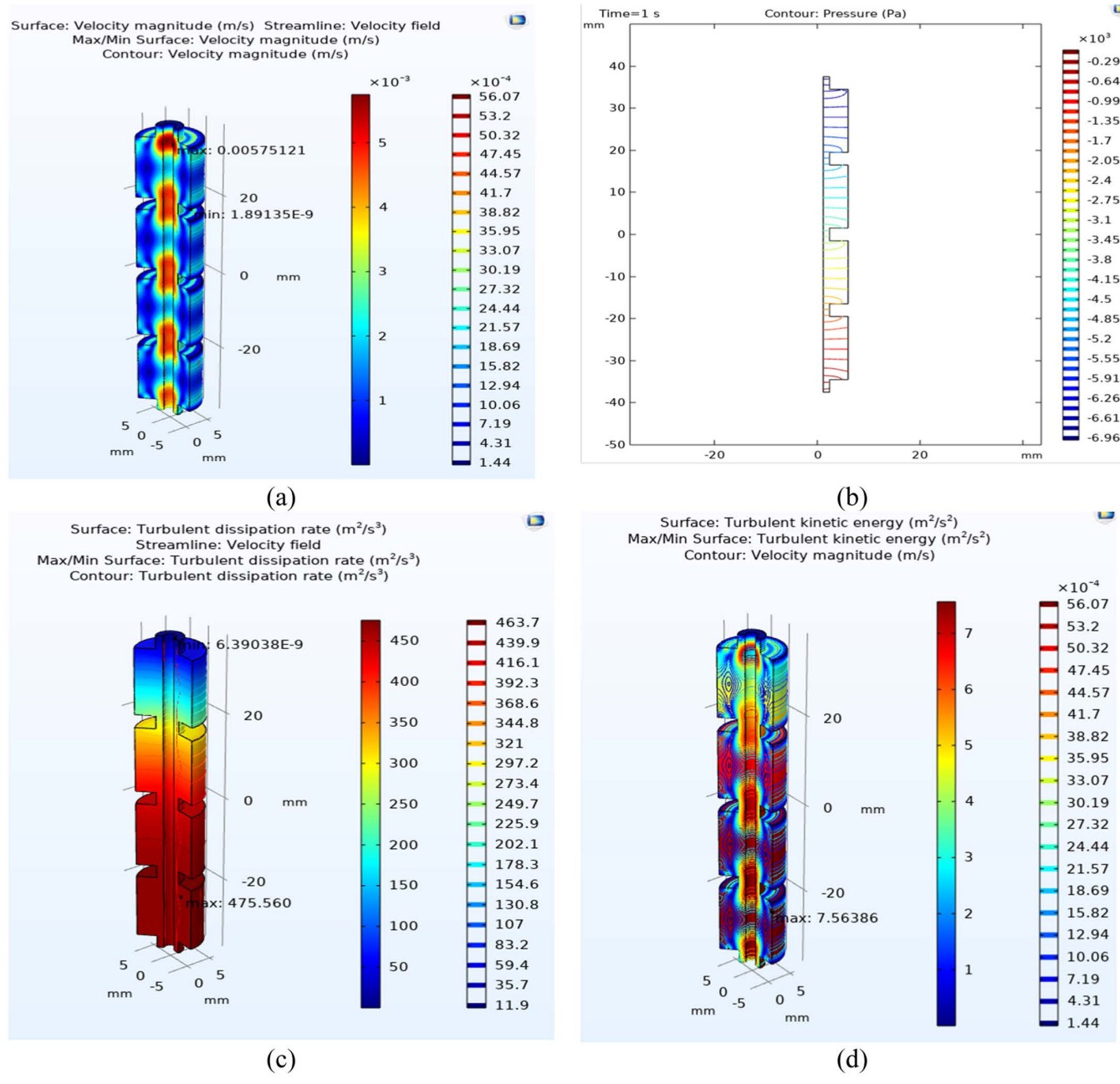

**Fig 10. Contour:** a) velocity, b) pressure, c) turbulent kinetic energy (TKE), and d) turbulent energy dissipation of optimal conditions.

other tested ranges by achieving a high yield with a 20% reduction in energy dissipation compared to previous studies [10], making them ideal for sustainable biodiesel production.

### 3.9. Validation of fluid simulation

In order to produce biodiesel from sunflower oil, this work examined and adjusted TKE in various single-orifice reactor geometries. The average and maximum TKE values were determined for different reactor designs, including baffle

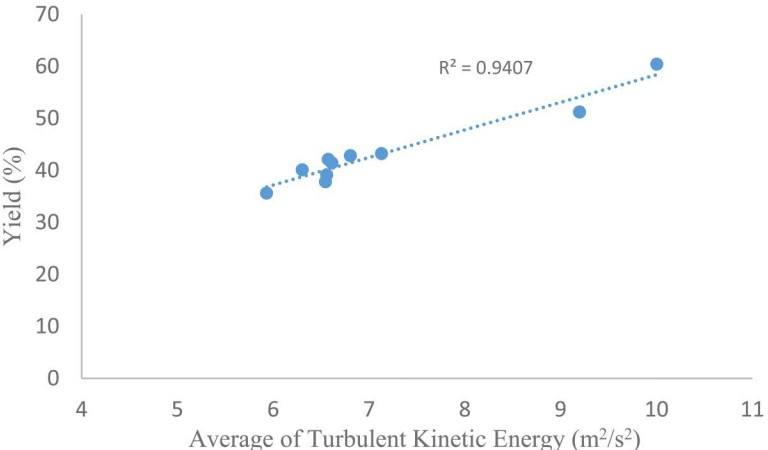

**Fig 11. Relationship between the average of Turbulent kinetic energy and biodiesel yield.**

dimensions, spacing, and oscillation frequency modifications, using computational simulations with the k-ε turbulent model. The findings demonstrated that improved reactor layouts greatly raised the intensity of turbulence, which is essential for improving mass transfer, mixing, and chemical reaction rates.

To verify the predictions, a laboratory experiment was conducted to produce biodiesel using sunflower oil. The biodiesel yield was calculated for various reactor geometries and contrasted with the predicted turbulent variables. The TKE average, TKE maximum, vorticity average, and vorticity maximum were shown to be correlated with biodiesel conversion at 0.972, 0.970, 0.882, and 0.735, respectively. This implies that greater turbulent levels, as higher TKE indicates, improve reactant mixing and accelerate reaction rates, increasing biodiesel output. This study demonstrates how crucial reactor design is to producing biodiesel effectively by enhancing turbulence. High frequencies increase energy use but significantly enhance mixing and yield (83%), offering clear advantages in production efficiency. Energy consumption, while not measured here, can be further optimized to maximize industrial benefits. The equation between the TKE average and biodiesel yield is represented in Fig 11.

## 4. Conclusion

Optimization of a single-orifice oscillatory flow reactor advances sustainable biodiesel production from sunflower oil. Simplified design achieves an 83% yield, with turbulent kinetic energy (TKE) of 7.56 m²/s², maximum vorticity of 112.23 1/s, energy dissipation of 359.82 m²/s³, and pressure difference of 7545.02 Pa, validated by a TKE-yield correlation ($R^2 = 0.972$). Compared to multi-orifice reactors (88% yield, higher costs) and smooth periodic constriction reactors (74.5% yield), energy dissipation reduces by 20% (359.82 m²/s³ vs. 655.06 m²/s³). Scalable, energy-efficient design minimizes carbon emissions and integrates into industrial biodiesel workflows. Findings offer a novel, cost-effective contribution to the knowledge base, enabling eco-friendly biofuel production with enhanced efficiency.

## Nomenclature

| CFD | Computational Fluid Dynamics |
|-----|------------------------------|
| RSM | Response Surface Method |
| TKE | Turbulent Kinetic Energy |
| OFR | Oscillatory Flow Reactor |
| KOH | Potassium hydroxide |

## Supporting information

**S1 Data.** **Raw data output from computer simulation.**
(XLSX)

## Acknowledgments

The authors would like to thank the Research Council of Shahrekord University for providing the necessary equipment and space to carry out this work.

## Author contributions

**Data curation:** Bahram Hosseinzadeh Samani.

**Formal analysis:** Sajad Rostami.

**Investigation:** Fatemeh Khadivi.

**Supervision:** Bahram Hosseinzadeh Samani.

**Validation:** Bahram Hosseinzadeh Samani.

**Writing – original draft:** Fatemeh Khadivi.

**Writing – review & editing:** Bahram Hosseinzadeh Samani, Kimia Taki, Mohammadreza Asghari, Shirin Ghatrehsamani.

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
