## [Decision Letter · Decision Letter 0]

23 Mar 2025

Dear Dr. Hosseinzadeh Samani,

Thank you for submitting your manuscript to PLOS ONE. After careful consideration, we feel that it has merit but does not fully meet PLOS ONE’s publication criteria as it currently stands. Therefore, we invite you to submit a revised version of the manuscript that addresses the points raised during the review process.

**ACADEMIC EDITOR: I suggest a major revision is required as per the comments made by the reviewers as below** :

We look forward to receiving your revised manuscript.

Kind regards,

Mohammad Yusuf, Ph.D.

Academic Editor

PLOS ONE

Journal Requirements:

The Research Council of Shahrekord University is thankfully acknowledged for the financial support to carry out the work (grant No:02GRN1M1796).

4. We note that your Data Availability Statement is currently as follows: All relevant data are within the manuscript and its Supporting Information files

Reviewers' comments:

Reviewer's Responses to Questions

**Comments to the Author**

1. Is the manuscript technically sound, and do the data support the conclusions?

Reviewer #1: Partly

Reviewer #2: Yes

Reviewer #3: Yes

2. Has the statistical analysis been performed appropriately and rigorously?

Reviewer #1: I Don't Know

Reviewer #2: Yes

Reviewer #3: Yes

3. Have the authors made all data underlying the findings in their manuscript fully available?

Reviewer #1: Yes

Reviewer #2: Yes

Reviewer #3: Yes

4. Is the manuscript presented in an intelligible fashion and written in standard English?

Reviewer #1: No

Reviewer #2: Yes

Reviewer #3: Yes

Reviewer #1: Reviewer Reports:

I recommend a major amendment at this level.

General comments:

The manuscript entitled “Optimization and CFD-RSM Analysis of Single Orifice Reactor for Enhanced Biodiesel Production” was reviewed. The work carried out in the manuscript is interesting and aimed at optimizing and analyzing a single-orifice oscillatory flow reactor using the Computational Fluid Dynamics (CFD)-Response Surface Method (RSM) to enhance biodiesel production from sunflower oil. Better connect your research findings to previous works published in PONE and in other top journals. Please also remove ANY lumped references. Please define each of them separately to avoid inappropriate citations. It is recommended that the authors work with a science editor who is proficient in the native English language to improve the organization and delivery of some portions of the manuscript. Too many abbreviations are used in the analysis and results. I recommend a nomenclature section for the abbreviations and variables used throughout the passage. The journal's author guidelines and instructions should be followed in preparing the revised version. Other main remarks that, in my opinion, need attention are the following:

Detailed comments:

The abstract should state briefly the purpose of the research, the principal results, and major conclusions. In the abstract, please add an indication of the achievements from your study that are relevant to the journal's scope. Please be concise—maximum 1-2 lines.

The review of the literature needs more updating with works to have a clear and concise state-of-the-art analysis. This should more clearly show the knowledge gaps identified and link them to the paper's goals. While the general introduction is acceptable, the state-of-the-art review that follows is very difficult to understand, and no specific thoughts can be inferred. The major defect of this study is the debate or argument is not clearly stated. The relevant reference may be of interest to the author according to below: Development of Heterogeneous/Nanocatalysts in Biodiesel Production; Magnetic graphene oxide supported tin oxide (SnO) nanocomposite as a heterogeneous catalyst for biodiesel production from soybean oil; Exploring Waste-Derived Catalysts for Sustainable Biodiesel Production: A Path Towards Renewable Energy; Please eliminate the use of redundant words. Eg. In this way, Recently, Respectively, therefore, currently, thus, hence, finally, to do this, first, in order, however, moreover, nowadays, today, consequently, in addition, additionally, furthermore. Please revise all similar cases, as removing these term(s) would not significantly affect the meaning of the sentence. This will keep the manuscript as CONCISE as possible. Please check ALL. Avoid beginning or ending a sentence with one or a few words; they are usually redundant. Kindly revise all.

Some sections could benefit from clearer explanations, especially the methodology. Please avoid having one heading after another with no discussion in between, as in the case of Sections 2 and 2.1. Kindly inspect the entire document for similar instances and revise accordingly. Please add in the beginning your scientific hypothesis. In the course of describing the performed actions, please provide reader guidance, sufficient for understanding why those actions have been performed. The percentage purity and company of all reagents/chemicals utilized must be reported. Provide more detail on the CFD parameters used. For instance, what turbulence model was applied, and why was it chosen? Clarify the selection and optimization process for the reactor parameters. A flowchart might help enhance understanding.

This is the weakest section of the entire manuscript. All the findings of the current work need to be compared and discussed with the results of other researchers finding instead of having a general comparison with other researchers' works. The authors should perform a comparison between the forecasting results. In your discussion section, please link your empirical results with a broader and deeper literature review. The figures and tables are adequately presented, but some graphs lack adequate labeling or legends, making it difficult to interpret the data fully. While the authors mention experimental validation of their simulations, further details on the experimental setup, conditions, and results would strengthen the manuscript. The discussion provides a good overview of findings, but it could be enhanced by comparing results with previous studies in the field. This would contextualize the significance of the findings and demonstrate their contribution to the field.

Please make sure your conclusions section underscores the scientific value-added of your paper, and/or the applicability of your findings/results. Highlights the novelty of your study. In the conclusions, in addition to summarising the actions taken and results, please strengthen the explanation of their significance. It is recommended to use quantitative reasoning compared with appropriate benchmarks, especially those stemming from previous work.

Ensure all cited works are up to date and relevant. Some references are older; consider including more recent studies to enhance the literature review.

Reviewer #2: Manuscript Review: Optimization and CFD-RSM Analysis of Single Orifice Reactor for Enhanced Biodiesel Production

1. Key Scientific Contributions and Strengths

This study presents a well-structured approach to optimizing a single-orifice oscillatory flow reactor (OFR) for biodiesel production, utilizing Computational Fluid Dynamics (CFD) and Response Surface Methodology (RSM). The combination of fluid dynamic modeling and experimental validation is a strong aspect, particularly the attempt to correlate Turbulent Kinetic Energy (TKE), vorticity, and biodiesel yield. One of the major highlights is that the optimized reactor conditions (12.12 Hz frequency, d₀/D = 0.4-, and 10 mm baffle spacing) led to an 83% biodiesel conversion, showing the potential of oscillatory reactors for efficient mixing and mass transfer. The methodology, which involves varying baffle diameter ratio, frequency, and spacing, then using RSM to derive optimal conditions, is sound. By sharing their observation on the effects of pressure drop, vorticity, and turbulence intensity, the study contributes to efficient design of reactors for biodiesel yield.

2. Revisions/clarifications required

2.1. Lack of Clear Justification for Optimal Conditions

The manuscript states that the optimized reactor parameters were a frequency of 12.12 Hz, a baffle diameter ratio (d₀/D) of 0.4, and a baffle spacing of 10 mm (Abstract). However, it does not clearly explain why these conditions were chosen over others within the tested range. In Table 2, the study varied d₀/D between 0.4 - 0.8, frequency between 5 - 15 Hz, and baffle spacing between 3 - 7 mm. But the manuscript does not explicitly state how biodiesel yield varied with these parameters, making it unclear whether these values were chosen purely based on TKE, vorticity, or a trade-off with biodiesel yield.

2.2. Insufficient Discussion on Practical Limitations and Industrial Feasibility

The study convincingly shows that higher frequency and increased turbulence enhance biodiesel yield, but it does not discuss real-world operational limitations of oscillatory flow reactors. High-frequency oscillations may improve mixing, but they also increase energy consumption. There is no mention of whether the energy cost associated with high oscillation frequencies offsets the gain in biodiesel yield.

2.3. CFD Model Limitations and Assumptions

The manuscript models the reactor using a single-phase flow assumption with the k-ε turbulence model (Section 2.3.1). While k-ε is widely used, it struggles with near-wall turbulence predictions and may underpredict the effects of flow separation around baffles. The authors should briefly explain why they did not use a k-ω SST model which would provide a more accurate representation of turbulence and vortex shedding.

2.4. Experimental Validation Lacks GC Details

The experimental validation section (2.5) states that biodiesel yield was measured using Gas Chromatography (GC), but does not provide details such as: Gc sample preparation? the general settings of the GC? Specifications of the GC used? This will help with the reproducibility and comparison of results.

3. Declaration

I could not correlate the figure discussions to figures as the figures were not attached to the manuscript.

4. Recommendation

This manuscript presents a valuable contribution to the field of biodiesel reactor optimization and should be accepted for publication after the minor revisions are concisely done.

Reviewer #3: Paper Title: Optimization and CFD-RSM Analysis of Single Orifice Reactor for Enhanced Biodiesel Production

Journal Title: PLOS ONE

Manuscript No.: PONE-D-25-06866

Article type: Research Article

The present investigation, titled “Optimization and CFD-RSM Analysis of Single Orifice Reactors for Enhanced Biodiesel Production,” demonstrates the potential of single orifice reactors to enhance biodiesel production through improved turbulance and mixing efficiency. Further, the optimal dimensions of the reactor design were determined using numerical simulations and experimental analysis. role of O-H bond scission in ethanol steam reforming is emphasized. The topic is relevant given the increasing global interest in sustainable energy solutions and the pivotal role that biodiesel is expected to play in the transition towards a clean energy future. However, there are a few areas where the manuscript could be further improved with minor revisions:

1. Design the Abstract part, and it must be more informative by including more mathematical findings and more powerful explanations (For e.g., also discuss the parameters effect on biodiesel yield). Please be concise—maximum 1-2 lines.

2. The author should use new and more references in the introduction (2024, 2023, 2022 , …).

3. For consistency, write the abbreviation of the terminology and use these abbreviations throughout the manuscript to avoid repetition of terminologies [e.g., turbulent kinetic energy (TKE), computational fluid dynamics (CFD), etc…].

4. In between the words, the capital words have been used. Please revise these.

5. In the result discussion section, the authors discuss the result but mention a few outdated references. Please discuss your results with previous recent studies.

6. Can you explain the statistical procedures used to ensure reliable and accurate findings in the RSM modeling?

7. Citations are included but could be better integrated into the narrative. Some sections might benefit from additional references to support claims or provide further reading.

8. The author should consider adding references from the past few years (2020 onward) to reflect the latest developments in the field.

9. In Table 1, author, please check the values of “the ratio of baffle diameter to reactor diameter” (For e.g., it is 0.4-0.8 or 0/4-0/8)?

10. Please check all the references cited in the text carefully and correct the inconsistency (For e.g., Line 554, check the reference format, i.e., Sreenivasan & Vedantam, 2024).

11. A professional proofread should be conducted again for the whole manuscript due to many typos.

**Do you want your identity to be public for this peer review?** For information about this choice, including consent withdrawal, please see our Privacy Policy

Reviewer #1: No

Reviewer #2: No

Reviewer #3: No

---

## [Author Response · Author response to Decision Letter 1]

9 Apr 2025

Response to the honorable referee of the paper entitled:

Optimization and CFD-RSM Analysis of Single Orifice Reactor for Enhanced Biodiesel Production

Thanking the comments and proposed amendments of the honorable referees, the answer to the referred points is mentioned in the separation of different parts of the paper as follows:

Reviewer #1

1. Better connect your research findings to previous works published in Plos One and in other top journals.

Response: Thanks for your attention. It was done.

2. Please remove ANY lumped references. Define each of them separately to avoid inappropriate citations.

Response: Thanks for your attention. It was corrected in the text.

3. It is recommended that the authors work with a science editor who is proficient in the native English language to improve the organization and delivery of some portions of the manuscript.

Response: Thanks for your recommendation. It was done.

4. Too many abbreviations are used in the analysis and results. I recommend a nomenclature section for the abbreviations and variables used throughout the passage.

Response: Thanks for mentioning that. It was prepared and added between the abstract and introduction sections.

5. The journal's author guidelines and instructions should be followed in preparing the revised version.

Response: Thanks for your comment. The manuscript was revised with plos one's author guidelines.

6. The abstract should state briefly the purpose of the research, the principal results, and major conclusions. In the abstract, please add an indication of the achievements from your study that are relevant to the journal's scope. Please be concise—maximum 1-2 lines.

Response: Thanks for mentioning that. The revised abstract now includes purpose, results, and conclusions with a brief note on practical sustainability benefits.

7. The review of the literature needs more updating with works to have a clear and concise state-of-the-art analysis. This should more clearly show the knowledge gaps identified and link them to the paper's goals.

Response: Thanks for your suggestion. The literature review in introduction section was updated with recent studies.

8. The major defect of this study is the debate or argument is not clearly stated.

Response: Thanks for mentioning that. As it was added to the end of the introduction section, the aim of this study was to design and simulate a single-orifice oscillatory reactor and analyze the turbulent kinetic energy, vorticity and dissipation factors inside this reactor to increase mixing and also increase the yield of biodiesel. In this study, the optimal reactor design dimensions were determined using numerical simulation and experimental analysis.

9. Please eliminate the use of redundant words. Eg. In this way, Recently, Respectively, therefore, currently, thus, hence, finally, to do this, first, in order, however, moreover, nowadays, today, consequently, in addition, additionally, furthermore. Please revise all similar cases, as removing these term(s) would not significantly affect the meaning of the sentence. This will keep the manuscript as CONCISE as possible. Please check ALL.

Response: Thanks for your suggestion. It was corrected in the text.

10. Avoid beginning or ending a sentence with one or a few words; they are usually redundant. Kindly revise all.

Response: Thanks for your suggestion. It was revised.

11. Some sections could benefit from clearer explanations, especially the methodology. Please avoid having one heading after another with no discussion in between, as in the case of Sections 2 and 2.1. Kindly inspect the entire document for similar instances and revise accordingly.

Response: Thanks for your suggestion. A brief explanation was added between sections 2 and 2.1.

12. Please add in the beginning your scientific hypothesis.

Response: Thanks for your suggestion. It was added at the end of the introduction section.

13. In the course of describing the performed actions, please provide reader guidance, sufficient for understanding why those actions have been performed.

Response: Thanks for your benefit comment. it was added at the beginning of each subsection in the materials and methods section. And also, it was added in flowchart (figure 1).

14. The percentage purity and company of all reagents/chemicals utilized must be reported.

Response: Thanks for your attention. It was added in section (2.5).

15. Provide more detail on the CFD parameters used. For instance, what turbulence model was applied, and why was it chosen?

Response: Thanks for mentioning that. It was added in section (2.3.1).

16. Clarify the selection and optimization process for the reactor parameters. A flowchart might help enhance understanding.

Response: Thanks for your suggestion. A flowchart (figure 1) was added to the beginning of the materials and methods section.

17. All the findings of the current work need to be compared and discussed with the results of other researchers finding instead of having a general comparison with other researchers' works. The authors should perform a comparison between the forecasting results.

Response: Thanks for your suggestion. It was added in the result and discussion section.

18. In your discussion section, please link your empirical results with a broader and deeper literature review.

Response: Thanks for your suggestion. It was added in the result and discussion section.

19. The figures and tables are adequately presented, but some graphs lack adequate labeling or legends, making it difficult to interpret the data fully.

Response: Thanks for your attention. It was corrected in the manuscript.

20. While the authors mention experimental validation of their simulations, further details on the experimental setup, conditions, and results would strengthen the manuscript.

Response: Thanks for your suggestion. It was corrected in the text

21. The discussion provides a good overview of findings, but it could be enhanced by comparing results with previous studies in the field. This would contextualize the significance of the findings and demonstrate their contribution to the field.

Response: Thanks for your suggestion. It was added in the result and discussion section.

22. Please make sure your conclusions section underscores the scientific value-added of your paper, and/or the applicability of your findings/results.

Response: Thanks, it was revised.

23. Highlights the novelty of your study in conclusion section.

Response: Thanks, it was done.

24. In the conclusions, in addition to summarizing the actions taken and results, please strengthen the explanation of their significance. It is recommended to use quantitative reasoning compared with appropriate benchmarks, especially those stemming from previous work.

Response: Thanks for your benefit comment. The conclusion was revised.

25. Ensure all cited works are up to date and relevant. Some references are older; consider including more recent studies to enhance the literature review.

Response: Thanks, it was done.

Reviewer #2

1. The manuscript states that the optimized reactor parameters were a frequency of 12.12 Hz, a baffle diameter ratio (d₀/D) of 0.4, and a baffle spacing of 10 mm (Abstract). However, it does not clearly explain why these conditions were chosen over others within the tested range. In Table 2, the study varied d₀/D between 0.4 - 0.8, frequency between 5 - 15 Hz, and baffle spacing between 3 - 7 mm. But the manuscript does not explicitly state how biodiesel yield varied with these parameters, making it unclear whether these values were chosen purely based on TKE, vorticity, or a trade-off with biodiesel yield.

Response: Thanks for your comment. A flowchart was added to section (2) to clarify the process. And also, some explanations were added to section (3.8). as it shown ranges for d₀/D, frequency, and baffle spacing (Table 2) were derived from Section (2.2) equations, then used in RSM to generate different options for reactor designs. These were simulated using CFD methods to obtain TKE, dissipations, and vorticity, which RSM analyzed to find the optimal point. Experiments validated this with an 83% biodiesel yield, linked to improved mixing. Among the tested ranges, these conditions outperformed others by balancing high TKE and vorticity with lower energy dissipation leading to the highest biodiesel yield.

2. The study convincingly shows that higher frequency and increased turbulence enhance biodiesel yield, but it does not discuss real-world operational limitations of oscillatory flow reactors.

High-frequency oscillations may improve mixing, but they also increase energy consumption. There is no mention of whether the energy cost associated with high oscillation frequencies offsets the gain in biodiesel yield.

Response: Thank you for your benefit comment. we have added a discussion in section (3.9) addressing the practical limitations of high-frequency oscillations. While increased frequencies enhance mixing and biodiesel yield, the energy consumption also increased duo to the greater oscillatory power requirements. Although specific energy costs were not quantified in this study, the significant yield improvement suggests a potential efficiency gain, warranting further investigation into energy trade-offs for industrial feasibility.

3. The manuscript models the reactor using a single-phase flow assumption with the k-ε turbulence model (Section 2.3.1). While k-ε is widely used, it struggles with near-wall turbulence predictions and may underpredict the effects of flow separation around baffles.

The authors should briefly explain why they did not use a k-ω SST model which would provide a more accurate representation of turbulence and vortex shedding.

Response: Thanks for mentioning that. It was added in section (2.3.1). The main reason for choosing the k- ε model for CFD simulation is its efficiency and reliability in simulating the intense turbulent flows of an oscillatory reactor. It excels at capturing TKE and its dissipation rate, crucial for enhancing mixing in biodiesel production. While models like k-ω SST offer precision near walls, k-ε strikes an effective balance between accuracy and computational simplicity, ideal for focus on bulk flow dynamics.

4. The experimental validation section (2.5) states that biodiesel yield was measured using Gas Chromatography (GC), but does not provide details such as: GC sample preparation? The general settings of the GC? Specifications of the GC used? This will help with the reproducibility and comparison of results.

Response: Thanks for mentioning that. It was added in section (2.5.1).

5. I could not correlate the figure discussions to figures as the figures were not attached to the manuscript.

Response: Thanks for noting this issue. The figures were submitted in a separate file labeled "Figures" with the original manuscript.

Reviewer #3

1. Design the Abstract part, and it must be more informative by including more mathematical findings and more powerful explanations (For e.g., also discuss the parameters effect on biodiesel yield). Please be concise—maximum 1-2 lines.

Response: Thanks for your attention. corrected in the text.

2. The author should use new and more references in the introduction (2024, 2023, 2022, …).

Response: Thanks for your suggestion. corrected in the text.

3. For consistency, write the abbreviation of the terminology and use these abbreviations throughout the manuscript to avoid repetition of terminologies [e.g., turbulent kinetic energy (TKE), computational fluid dynamics (CFD), etc…].

Response: Thanks for your suggestion. It was added between abstract and introduction section.

4. In between the words, the capital words have been used. Please revise these.

Response: Thanks for your attention. It was corrected in the text.

5. In the result and discussion section, the authors discuss the result but mention a few outdated references. Please discuss your results with previous recent studies.

Response: Thanks for your attention. It was added in results and discussion section.

6. Can you explain the statistical procedures used to ensure reliable and accurate findings in the RSM modeling?

Response: Thank you for your question. Explanation was added to the different part of section (2). As it shown in section (2) ranges for d₀/D, frequency, and baffle spacing (Table 2) were derived, then used in RSM to generate 29 runs from a Box-Behnken design for reactor designs then simulating them with using CFD methods to obtain the outputs (TKE, dissipations, and vorticity), and then start optimizing based on outputs. The optimal condition simulated using CFD and the outputs compare with the optimal point of RSM. By comparing these two we understand that the optimal condition matched CFD results and experimental yield (83%), confirming robustness.

7. Citations are included but could be better integrated into the narrative. Some sections might benefit from additional references to support claims or provide further reading.

Response: Thanks for your attention. corrected in the text.

8. The author should consider adding references from the past few years (2020 onward) to reflect the latest developments in the field.

Response: Thanks for your attention. It was done.

9. In Table 1, author, please check the values of “the ratio of baffle diameter to reactor diameter” (For e.g., it is 0.4-0.8 or 0/4-0/8)?

Response: Thanks for mentioning that. It was corrected in the text.

10. Please check all the references cited in the text carefully and correct the inconsistency (For e.g., Line 554, check the reference format, i.e., Sreenivasan & Vedantam, 2024).

Response: Thanks for your attention. It was corrected in the text.

11. A professional proofread should be conducted again for the whole manuscript due to many typos.

Response: Thanks for your attention. It was revised.

---

## [Decision Letter · Decision Letter 1]

13 May 2025

Dear Dr. Hosseinzadeh Samani,

Thank you for submitting your manuscript to PLOS ONE. After careful consideration, we feel that it has merit but does not fully meet PLOS ONE’s publication criteria as it currently stands. Therefore, we invite you to submit a revised version of the manuscript that addresses the points raised during the review process.

We look forward to receiving your revised manuscript.

Kind regards,

Mohammad Yusuf, Ph.D.

Academic Editor

PLOS ONE

Additional Editor Comments:

After revieweing the review reports of all the reviewers; I recommend major revision at this stage.

The author is suggested to re-check and address the queries that are raised by the reviewer in detail as per reviewer's suggestion.

Reviewers' comments:

Reviewer's Responses to Questions

**Comments to the Author**

Reviewer #1: (No Response)

Reviewer #2: All comments have been addressed

Reviewer #3: All comments have been addressed

2. Is the manuscript technically sound, and do the data support the conclusions?

Reviewer #1: Partly

Reviewer #2: Yes

Reviewer #3: Yes

3. Has the statistical analysis been performed appropriately and rigorously?

Reviewer #1: No

Reviewer #2: Yes

Reviewer #3: Yes

4. Have the authors made all data underlying the findings in their manuscript fully available?

Reviewer #1: Yes

Reviewer #2: Yes

Reviewer #3: Yes

5. Is the manuscript presented in an intelligible fashion and written in standard English?

Reviewer #1: No

Reviewer #2: Yes

Reviewer #3: Yes

Reviewer #1: Reviewer Reports:

I have reviewed the revised version of the manuscript entitled "Optimization and CFD-RSM Analysis of Single Orifice Reactor for Enhanced Biodiesel Production". The paper has been improved only in a few sections and can not be accepted at this stage. I appreciate your effort to improve your manuscript; however, deeper corrections would be required before its publication. Please, one more time, check all my previous comments and submit them with a highlights version. Also, give the accurate location in the revised version. The methods are not adequately explained. The parameter of enhancement rate does not add value to the discussion. The discussion of the results is vague. It is better not to use the first person's pronoun. Do not use "we, us, or our" throughout the paper. Redaction and English grammar need to be revised. Some areas still require clarification, strengthening, and additional context to fully appreciate the novelty and robustness of the work.

Please also remove ANY lumped references. Please define each of them separately to avoid inappropriate citations.

What is the unique contribution of this paper? The indication of this feature should start from the abstract. Otherwise, not too many readers will bother reading the paper after looking at the abstract. The abstract does not work well. It narrates activities. A good abstract should address these issues: what are you trying to do, why, what you found, and what is the significance of your findings.

The introduction section should follow the state of the art of this field and review what has been done to support the research gap and the significance of this study. Please improve the state-of-the-art overview to clearly show the progress beyond the state of the art. The lack of proper justification creates the wrong impression that the authors are unaware of the recent developments. These are a few good examples for your reference: Development of Heterogeneous/Nanocatalysts in Biodiesel Production; Magnetic graphene oxide supported tin oxide (SnO) nanocomposite as a heterogeneous catalyst for biodiesel production from soybean oil; Exploring Waste-Derived Catalysts for Sustainable Biodiesel Production: A Path Towards Renewable Energy; Please improve the aim of the introduction.

Expand the description of the experimental setup, including specifics such as reactor dimensions, operating conditions, and chemical reagent purity. Providing quantitative comparisons between CFD predictions and experimental results (biodiesel yield, reaction conversion rates) with statistical analysis will substantiate the validity of the simulation approach.

While the optimised conditions are well-presented, the manuscript should explicitly clarify how the specific values (frequency of 12.12 Hz, d₀/D of 0.4, and baffle spacing of 10 mm) were selected from the tested ranges. Including a brief discussion on the trade-offs between turbulence metrics (TKE, vorticity) and biodiesel yield, supported by data or graphs, will strengthen the rationale behind these choices.

The discussion section would benefit from a more detailed comparison of your results with recent studies in similar reactor designs or biodiesel production processes. Highlighting how your optimized parameters outperform or align with existing literature will better position your findings within the current scientific landscape. Specify the turbulence model used (k-ε, k-ω, LES), along with justification for its selection. Discuss how mesh independence and boundary conditions were determined to ensure simulation accuracy.

The conclusion is pretty generic and fails to provide any improvement in the existing knowledge base. The conclusion should be concise and to the point, indicating the application of the work. Please make sure your conclusions section underscores the scientific value-added of your paper, and/or the applicability of your findings/results. Highlight the novelty of your study. In the conclusion, emphasize the practical implications of your optimized reactor design, especially in terms of scalability, environmental benefits, and potential integration into existing biodiesel production workflows. Use quantitative benchmarks or comparisons to highlight the advances achieved.

Ensure all references are appropriately cited in the main text and that figure captions are consistently formatted.

Reviewer #2: The authors have thoroughly addressed the reviewers' comments and significantly improved their study. The manuscript is now well-suited for publication and represents a valuable contribution to the body of research in this area.

Reviewer #3: (No Response)

**Do you want your identity to be public for this peer review?** For information about this choice, including consent withdrawal, please see our Privacy Policy

Reviewer #1: No

Reviewer #2: No

Reviewer #3: No

---

## [Author Response · Author response to Decision Letter 2]

16 May 2025

Response to the honorable referee of the paper entitled:

Optimization and CFD-RSM Analysis of Single Orifice Reactor for Enhanced Biodiesel Production

Thanking the comments and proposed amendments of the honorable referees, the answer to the referred points is mentioned in the separation of different parts of the paper as follows:

Reviewer #1

1. The manuscript has improved in only a few sections and requires deeper corrections before publication. Re-check all previous comments and submit a highlights version, specifying the accurate location of changes in the revised version.

Response: Thanks for your feedback. All previous comments are checked and the changes are added in the revised manuscript.

2. The methods section is not adequately explained. Provide a more detailed description of the experimental setup, including reactor dimensions, operating conditions, and chemical reagent purity.

Response: Thanks for your attention. It was corrected in text.

3. The parameter of enhancement rate does not add value to the discussion and should be reconsidered or removed.

Response: Thanks for your recommendation. It was removed.

4. The discussion of the results is vague and needs to be more detailed. Include a comparison with recent studies on similar reactor designs or biodiesel production processes to better position the findings.

Response: Thanks for pointing that out. It was corrected in text.

5. Avoid using first-person pronouns ("we," "us," "our") throughout the paper to maintain a formal tone.

Response: Thanks for pointing that out. It was corrected in the text.

6. Redaction and English grammar need revision. Some sentences require clarification and strengthening to improve readability and professionalism.

Response: Thanks for your feedback. It was revised.

7. Remove lumped references and define each citation separately to avoid inappropriate citations.

Response: Thanks for your suggestion. It was corrected in text.

8. The unique contribution of the paper is unclear. Highlight this in the abstract, which currently narrates activities rather than addressing key points: what the study aims to do, why, what was found, and the significance of the findings.

Response: Thanks for your suggestion. The abstract of this paper was revised.

9. The introduction lacks a proper state-of-the-art overview. Review recent developments in the field (e.g., studies like "Development of Heterogeneous/Nanocatalysts in Biodiesel Production," "Magnetic graphene oxide supported tin oxide (SnO) nanocomposite as a heterogeneous catalyst for biodiesel production," "Exploring Waste-Derived Catalysts for Sustainable Biodiesel Production") to support the research gap and significance. Improve the aim of the introduction to show progress beyond the state of the art.

Response: Thank you for your valuable suggestion. The references have been incorporated in lines (102-107).

10. Provide quantitative comparisons between CFD predictions and experimental results (e.g., biodiesel yield, reaction conversion rates) with statistical analysis to validate the simulation approach.

Response: Thanks for your suggestion. we have added a detailed comparison in Section 3.8, lines (773-783).

11. Clarify how the optimized conditions (frequency of 12.12 Hz, d₀/D of 0.4, baffle spacing of 10 mm) were selected from the tested ranges. Discuss trade-offs between turbulence metrics (TKE, vorticity) and biodiesel yield, supported by data or graphs.

Response: Thanks for your comment. A flowchart was added to section (2) to clarify the process. And also, some explanations were added to section (3.8). as it shown ranges for d₀/D, frequency, and baffle spacing (Table 2) were derived from Section (2.2) equations, then used in RSM to generate different options for reactor designs. These were simulated using CFD methods to obtain TKE, dissipations, and vorticity, which RSM analyzed to find the optimal point. Experiments validated this with an 83% biodiesel yield, linked to improved mixing. Among the tested ranges, these conditions outperformed others by balancing high TKE and vorticity with lower energy dissipation leading to the highest biodiesel yield. Also, we have revised Section 3.8 (Process Optimization) to include a detailed explanation of how the specific values were determined using the Response Surface Method (RSM) and the trade-offs considered.

12. Specify the turbulence model used (k-ε, k-ω, LES) in the CFD simulations, with justification for its selection. Explain how mesh independence and boundary conditions were determined to ensure simulation accuracy.

Response: Thanks for your suggestion. The manuscript already specifies the use of the k-ε turbulence model in Section 2.3.1 (K-ε Turbulent Model) in lines (205-215), but we have expanded this section to include a more detailed justification for choosing the k-ε model over other models such as k-ω or Large Eddy Simulation (LES).

The importance of discussing mesh independence and boundary conditions were acknowledged to ensure the reliability of our CFD simulations. To address this, new subsection was added in Section 2.3 (in lines 180-197) to detail the mesh independence study and boundary condition determination.

13. The conclusion is generic and does not contribute to the existing knowledge base. Revise it to be concise, emphasizing the scientific value, novelty, practical implications (e.g., scalability, environmental benefits, integration into biodiesel production workflows), and quantitative benchmarks or comparisons.

Response: Thanks for your suggestion. The conclusion section was revised.

14. Ensure all references are appropriately cited in the main text and that figure captions are consistently formatted.

Response: Thanks for your attention. The references have been checked, additionally, captions of figures (1) and (10) were corrected in the manuscript and also in the attachment file.

---

## [Decision Letter · Decision Letter 2]

5 June 2025

Optimization and CFD-RSM Analysis of Single Orifice Reactor for Enhanced Biodiesel Production

PONE-D-25-06866R2

Dear Dr. Hosseinzadeh Samani,

We’re pleased to inform you that your manuscript has been judged scientifically suitable for publication and will be formally accepted for publication once it meets all outstanding technical requirements.

Kind regards,

Mohammad Yusuf, Ph.D.

Academic Editor

PLOS ONE

Additional Editor Comments (optional):

Reviewers' comments:

Reviewer's Responses to Questions

**Comments to the Author**

Reviewer #1: All comments have been addressed

2. Is the manuscript technically sound, and do the data support the conclusions?

Reviewer #1: Yes

3. Has the statistical analysis been performed appropriately and rigorously?

Reviewer #1: Yes

4. Have the authors made all data underlying the findings in their manuscript fully available?

Reviewer #1: Yes

5. Is the manuscript presented in an intelligible fashion and written in standard English?

Reviewer #1: Yes

Reviewer #1: Reviewer Reports:

I reviewed revised version manuscript titled" Optimization and CFD-RSM Analysis of Single Orifice Reactor for Enhanced Biodiesel Production". The paper has been improved and can be accepted.

**Do you want your identity to be public for this peer review?** For information about this choice, including consent withdrawal, please see our Privacy Policy

Reviewer #1: No

---

## [Editor Report · Acceptance letter]

PONE-D-25-06866R2

PLOS ONE

Dear Dr. Hosseinzadeh Samani,

I'm pleased to inform you that your manuscript has been deemed suitable for publication in PLOS ONE. Congratulations! Your manuscript is now being handed over to our production team.

Kind regards,

on behalf of

Dr. Mohammad Yusuf

Academic Editor

PLOS ONE